# AROMA: Preserving Spatial Structure for Latent PDE Modeling with Local Neural Fields

**Louis Serrano**[1] *    **Thomas X Wang**[1]    **Etienne Le Naour**[1,2]

**Jean-Noël Vittaut** [3]    **Patrick Gallinari**[1,4]

[1]Sorbonne Université, CNRS, ISIR, 75005 Paris, France
[2]EDF R&D, Palaiseau, France
[3]Sorbonne Université, CNRS, LIP6, 75005 Paris, France
[4]Criteo AI Lab, Paris, France

## Abstract

We present AROMA (Attentive Reduced Order Model with Attention), a framework designed to enhance the modeling of partial differential equations (PDEs) using local neural fields. Our flexible encoder-decoder architecture can obtain smooth latent representations of spatial physical fields from a variety of data types, including irregular-grid inputs and point clouds. This versatility eliminates the need for patching and allows efficient processing of diverse geometries. The sequential nature of our latent representation can be interpreted spatially and permits the use of a conditional transformer for modeling the temporal dynamics of PDEs. By employing a diffusion-based formulation, we achieve greater stability and enable longer rollouts compared to conventional MSE training. AROMA's superior performance in simulating 1D and 2D equations underscores the efficacy of our approach in capturing complex dynamical behaviors. *Github page*: https://github.com/LouisSerrano/aroma

## 1 Introduction

In recent years, many deep learning (DL) surrogate models have been introduced to approximate solutions to partial differential equations (PDEs) (Lu et al., 2021; Li et al., 2021; Brandstetter et al., 2022; Stachenfeld et al., 2022). Among these, the family of neural operators has been extensively adopted and tested across various scientific domains, demonstrating the potential of data-centric DL models in science (Pathak et al., 2022; Vinuesa & Brunton, 2022).

Neural Operators were initially constrained by discretization and domain geometry limitations. Recent advancements, such as neural fields (Yin et al., 2022; Serrano et al., 2023) and transformer architectures (Li et al., 2023; Hao et al., 2023), have partially addressed these issues, improving both dynamic modeling and steady-state settings. However, Neural Fields struggle to model spatial information and local dynamics effectively, and existing transformer architectures, while being flexible, are computationally expensive due to their operation in the original physical space and require large training datasets.

Our hypothesis is that considering spatiality is essential in modeling spatio-temporal phenomena, yet applying attention mechanisms directly is computationally expensive. We propose a new framework that models the dynamics in a reduced latent space, encoding spatial information compactly, by one or two orders of magnitude relative to the original space. This approach addresses both the complexity issues of transformer architectures and the spatiality challenges of Neural Fields.

---

*Corresponding author: louis.serrano@isir.upmc.fr

38th Conference on Neural Information Processing Systems (NeurIPS 2024).

Our novel framework leverages attention blocks and neural fields, resulting in a model that is easy to train and achieves state-of-the-art results on most datasets, particularly for complex geometries, without requiring prior feature engineering. To the best of our knowledge, we are the first to propose a fully attention-based architecture for processing domain geometries and unrolling dynamics. Compared to existing transformer architectures for PDEs, our framework first encapsulates the domain geometry and observation values in a compact latent representation, efficiently forecasting the dynamics at a lower computational cost. Transformer-based methods such as (Li et al., 2023; Hao et al., 2023) unroll the dynamics in the original space, leading to high complexity.

Our contributions are summarized as follows:

- A principled and versatile encode-process-decode framework for solving PDEs that operate on general input geometries, including point sets, grids, or meshes, and can be queried at any location within the spatial domain.

- A new spatial encode / process / decode approach: Variable-size inputs are mapped onto a fixed-size compact latent token space that encodes local spatial information. This latent representation is further processed by a transformer architecture that models the dynamics while exploiting spatial relations both at the local token level and globally across tokens. The decoding exploits a conditional neural field, allowing us to query forecast values at any point in the spatial domain of the equation.

- We include stochastic components at the encoding and processing levels to enhance stability and forecasting accuracy.

- Experiments performed on representative spatio-temporal forecasting problems demonstrate that AROMA is on par with or outperforms state-of-the-art baselines in terms of both accuracy and complexity.

## 2 Problem setting

In this paper, we focus on time-dependent PDEs defined over a spatial domain $\Omega$ (with boundary $\partial\Omega$) and temporal domain $[0, T]$. In the general form, their solutions $\boldsymbol{u}(x, t)$ satisfy the following constraints :

$$\frac{\partial \boldsymbol{u}}{\partial t} = F\left(\nu, t, x, \boldsymbol{u}, \frac{\partial \boldsymbol{u}}{\partial x}, \frac{\partial^2 \boldsymbol{u}}{\partial x^2}, \dots\right), \quad \forall x \in \Omega, \forall t \in (0, T] \tag{1}$$

$$\mathcal{B}(\boldsymbol{u})(t, x) = 0 \quad \forall x \in \partial\Omega, \forall t \in (0, T] \tag{2}$$

$$\boldsymbol{u}(0, x) = \boldsymbol{u}^0 \quad \forall x \in \Omega \tag{3}$$

where $\nu$ represents a set of PDE coefficients, Equations (2) and (3) represent the constraints with respect to the boundary and initial conditions. We aim to learn, using solutions data obtained with classical solvers, the evolution operator $\mathcal{G}$ that predicts the state of the system at the next time step: $\boldsymbol{u}^{t+\Delta t} = \mathcal{G}(\boldsymbol{u}^t)$. We have access to training trajectories obtained with different initial conditions, and we want to generate accurate trajectory rollouts for new initial conditions at test time. A rollout is obtained by the iterative application of the evolution operator $\boldsymbol{u}^{m\Delta t} = \mathcal{G}^m(\boldsymbol{u}^0)$.

## 3 Model Description

### 3.1 Model overview

We provide below an overview of the global framework and each component is described in a subsequent section. The model comprises three key components, as detailed in Figure 1.

- **Encoder** $\mathcal{E}_w : \boldsymbol{u}^t_{\mathcal{X}} \rightarrow \boldsymbol{Z}^t$. The encoder takes input values $\boldsymbol{u}^t_{\mathcal{X}}$ sampled over the domain $\Omega$ at time $t$, where $\mathcal{X}$ denotes the discrete sample space and could be a grid, an irregular mesh or a point set. $\boldsymbol{u}^t_{\mathcal{X}}$ is observed at locations $\boldsymbol{x} = (x_1, \dots x_N)$, with values $\boldsymbol{u}^t = (\boldsymbol{u}^t(\boldsymbol{x}_1), \cdots, \boldsymbol{u}^t(\boldsymbol{x}_N))$. $N$ is the number of observations and can vary across samples. $\boldsymbol{u}^t_{\mathcal{X}}$ is projected through a cross attention mechanism onto a set of $M$ tokens $\boldsymbol{Z}^t = (\boldsymbol{z}^t_1, \cdots, \boldsymbol{z}^t_M)$ with $M$ a fixed parameter. This allows

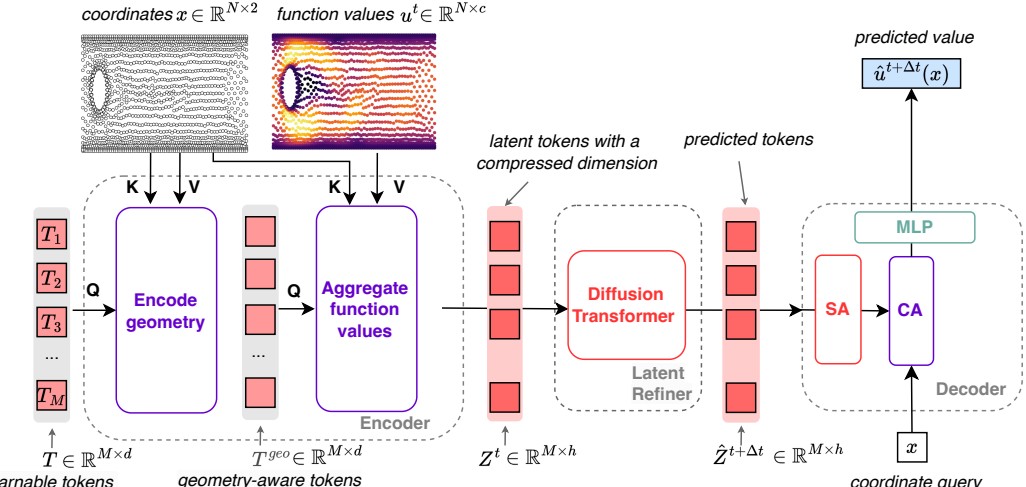

coordinates $x \in \mathbb{R}^{N \times 2}$  function values $u^t \in \mathbb{R}^{N \times c}$

latent tokens with a compressed dimension

predicted tokens

predicted value

$\hat{u}^{t+\Delta t}(x)$

MLP

K  V  K  V

$T_1$  $T_2$  $T_3$  ...  $T_M$

Q  Encode geometry  Q  Aggregate function values

Diffusion Transformer

SA  CA

Latent Refiner

Decoder

$x$

$T \in \mathbb{R}^{M \times d}$  $T^{geo} \in \mathbb{R}^{M \times d}$  $Z^t \in \mathbb{R}^{M \times h}$  $\hat{Z}^{t+\Delta t} \in \mathbb{R}^{M \times h}$

learnable tokens  geometry-aware tokens  coordinate query

Encoder

**Figure 1: AROMA inference**: The discretization-free encoder compresses the information of a set of $N$ input values to a sequence of $M$ latent tokens, where $M < N$. The conditional diffusion transformer is used to model the dynamics, acting as a latent refiner. The continuous decoder leverages self-attentions (SA), cross-attention (CA) and a local INR to map back to the physical space. Learnable tokens are shared and encode spatial relations. Latent token $Z^t$ represents $u_t$ and $Z^{t+\Delta t}$ is the prediction corresponding to $u_{t+\Delta t}$.

mapping any discretized input $u^t_{\mathcal{X}}$ onto a fixed dimensional latent representation $Z^t$ encoding implicit local spatial information from the input domain. The encoder is trained as a VAE and $Z^t$ is sampled from a multivariate normal statistics as detailed in Section 3.2.

- **Latent time-marching refiner** $\mathcal{R}_\theta : Z^t \to \hat{Z}^{t+\Delta t}$. We model the dynamics in the latent space through a transformer. The dynamics can be unrolled auto-regressively in the latent space for any time horizon without requiring to project back in the original domain $\Omega$. Self-attention operates on the latent tokens, which allows modeling global spatial relations between the local token representations. The transformer is enriched with a conditional diffusion mechanism operating between two successive time steps of the transformer. We experimentally observed that this probabilistic model was more robust than a baseline deterministic transformer for temporal extrapolation.

- **Decoder** $\mathcal{D}_\psi : \hat{Z}^{t+\Delta t} \to \hat{u}^{t+\Delta t}$. The decoder uses the latent tokens $\hat{Z}^{t+\Delta t}$ to approximate the function value $\hat{u}^{t+\Delta t}(x) = \mathcal{D}_\psi(x, \hat{Z}^{t+\Delta t})$ for any query coordinate $x \in \Omega$. We therefore denote $\hat{u}^{t+\Delta t} = \mathcal{D}_\psi(Z^{t+\Delta t})$ the predicted function.

**Inference**  We encode the initial condition and unroll the dynamics in the latent space by successive denoisings: $\hat{u}^{m\Delta t} = \mathcal{D}_\psi \circ \mathcal{R}_\theta^m \circ \mathcal{E}_w(u^0)$. We then decode along the trajectory to get the reconstructions. We outline the full inference pipeline in Figure 1 and detail its complexity analysis in Appendix C.1.

**Training**  We perform a two-stage training: we first train the encoder and decoder, secondly train the refiner. This is more stable than end-to-end training.

## 3.2  Encoder-decoder description

The encoder-decoder components are jointly trained using a VAE setting. The encoder is specifically designed to capture local input observation from any sampled point set in the spatial domain and encodes this information into a fixed number of tokens. The decoder can be queried at any position in the spatial domain, irrespective of the input sample.

**Encoder**  The encoder maps an arbitrary number $N$ of observations $(x, u(x)) := ((x_1, u(x_1)), \ldots, (x_N, u(x_N)))$ onto a latent representation $Z$ of fixed size $M$ through the following series of transformations:

$$\text{(i)} \quad (\boldsymbol{x}, \boldsymbol{u}(\boldsymbol{x})) \xrightarrow{\text{(positional, value) embeddings}} (\boldsymbol{\gamma}(\boldsymbol{x}), \boldsymbol{v}(\boldsymbol{x})) \in \mathbb{R}^{N \times d}$$

$$\text{(ii)} \quad (\mathbf{T}, \boldsymbol{\gamma}(\boldsymbol{x})) \xrightarrow{\text{geometry encoding}} \mathbf{T}^{\text{geo}} \in \mathbb{R}^{M \times d}$$

$$\text{(iii)} \quad (\mathbf{T}^{\text{geo}}, \boldsymbol{v}(\boldsymbol{x})) \xrightarrow{\text{observation spatial encoding}} \mathbf{T}^{\text{obs}} \in \mathbb{R}^{M \times d}$$

$$\text{(iv)} \quad \mathbf{T}^{\text{obs}} \xrightarrow{\text{dimension reduction}} \boldsymbol{Z} \in \mathbb{R}^{M \times h}$$

where $(\boldsymbol{\gamma}(\boldsymbol{x}), \boldsymbol{v}(\boldsymbol{x})) = ((\gamma(x_1), v(x_1)), \ldots, (\gamma(x_N), v(x_N)))$, and $h \ll d$.

**(i) Embed positions and observations**: Given an input sequence of coordinate-value pairs $(x_1, \boldsymbol{u}(x_1)), \ldots, (x_N, \boldsymbol{u}(x_N))$, we construct sequences of positional embeddings $\boldsymbol{\gamma} = (\gamma(x_1), \ldots, \gamma(x_N))$ and value embeddings $\boldsymbol{v} = (v(x_1), \ldots, v(x_N))$, where $\gamma(x) = \texttt{FourierFeatures}(x; \omega)$ and $v(x) = \texttt{Linear}(\boldsymbol{u}(x))$, with $\omega$ a fixed set of frequencies. These embeddings are aggregated onto a smaller set of learnable query tokens $\mathbf{T} = (T_1, \ldots, T_M)$ and then $\mathbf{T}' = (T_1', \ldots, T_M')$ with $M$ fixed, to compress the information and encode the geometry and spatial latent representations.

**(ii) Encode geometry**: Geometry-aware tokens $\mathbf{T}$ are obtained with a multihead cross-attention layer and a feedforward network (FFN), expressed as $\mathbf{T}^{\text{geo}} = \mathbf{T} + \texttt{FFN}(\texttt{CrossAttention}(\mathbf{Q} = \mathbf{W}_Q \mathbf{T}, \mathbf{K} = \mathbf{W}_K \boldsymbol{\gamma}, \mathbf{V} = \mathbf{W}_V \boldsymbol{\gamma}))$. This step does not include information on the observations, ensuring that similar geometries yield similar query tokens $\mathbf{T}^{\text{geo}}$ irrespective of the $\boldsymbol{u}$ values.

**(iii) Encode observations**: The $\mathbf{T}^{\text{geo}}$ tokens are then used to aggregate the observation values via a cross-attention mechanism: $\mathbf{T}^{\text{obs}} = \mathbf{T}^{\text{geo}} + \texttt{FFN}(\texttt{CrossAttention}(\mathbf{Q} = \mathbf{W}_Q' \mathbf{T}^{\text{geo}}, \mathbf{K} = \mathbf{W}_K' \boldsymbol{\gamma}, \mathbf{V} = \mathbf{W}_V' \boldsymbol{v}))$. Here, the values contain information on the observation values, and the keys contain information on the observation locations.

**(iv) Reduce channel dimension and sample $\boldsymbol{Z}$**: The information in the channel dimension of $\mathbf{T}'$ is compressed using a bottleneck linear layer. To avoid exploding variance in this compressed latent space, we regularize it with a penalty on the $L_2$ norm of the latent code $\|\boldsymbol{Z}\|^2$. Introducing stochasticity through a variational formulation further helps to regularize the auto-encoding and obtain smoother representations for the forecasting step. For this, we learn the components of a Gaussian multivariate distribution $\boldsymbol{\mu} = \texttt{Linear}(\mathbf{T}^{\text{obs}})$ and $\log(\boldsymbol{\sigma}) = \texttt{Linear}(\mathbf{T}^{\text{obs}})$ from which the final token embedding $\boldsymbol{Z}$ is sampled.

**Decoder** The decoder's role is to reconstruct $\hat{\boldsymbol{u}}^{t+\Delta t}$ from $\hat{\boldsymbol{Z}}^{t+\Delta t}$, see Figure 1. Since training is performed in two steps ("encode-decode" first and then "process"), the decoder is trained to reconstruct $\hat{\boldsymbol{u}}^t$ for input $\boldsymbol{u}^t$. One proceeds as follows. **(i) Increase channel dimensions and apply self-attention**: The decoder first lifts the latent tokens $\boldsymbol{Z}$ to a higher channel dimension (this is the reverse operation of the one performed by the encoder) and then apply several layers of self-attention to get tokens $\boldsymbol{Z}'$. **(ii) Cross-attend**: The decoder applies cross-attention to obtain feature vectors that depend on the query coordinate $x$, $(\mathbf{f}_q^u(x)) = \texttt{CrossAttention}(\mathbf{Q} = \mathbf{W}_Q(\gamma_q(x)), \mathbf{K} = \mathbf{W}_K \boldsymbol{Z}', \mathbf{V} = \mathbf{W}_V \boldsymbol{Z}')$, where $\gamma_q$ is a Fourier features embedding of bandwidth $\omega_q$. **(iii) Decode with MLP**: Finally, we use a small MLP to decode this feature vector and obtain the reconstruction $\hat{\boldsymbol{u}}(x) = \texttt{MLP}(\mathbf{f}_q^u(x))$. In contrast with existing neural field methods for dynamics modeling, the feature vector here is local. In practice, one uses multiple cross attentions to get feature vectors with different frequencies (see Appendix Figures 7 and 8 for further details).

**Training** The encoder and decoder are jointly optimized as a variational autoencoder (VAE) (Kingma & Welling, 2013) to minimize the following objective : $\mathcal{L} = \mathcal{L}_{\text{recon}} + \beta \cdot \mathcal{L}_{KL}$; where $\mathcal{L}_{\text{recon}} = \texttt{MSE}(u_{\mathcal{X}}^t, \hat{u}_{\mathcal{X}}^t)$ is the reconstruction loss between the input and the reconstruction $\mathcal{D}_\psi(\boldsymbol{Z}^t, \mathcal{X})$ on the grid $\mathcal{X}$, with $\boldsymbol{Z}^t \sim \mathcal{N}(\boldsymbol{\mu^t}, (\boldsymbol{\sigma^t})^2)$ and $\boldsymbol{\mu}^t, \boldsymbol{\sigma}^t = \mathcal{E}_w(u_{\mathcal{X}}^t)$. The KL divergence loss $\mathcal{L}_{\text{KL}} = D_{\text{KL}}(\mathcal{N}(\boldsymbol{\mu}^t, (\boldsymbol{\sigma}^t)^2) \,\|\, \mathcal{N}(0, I))$ helps regularize the network and prevents overfitting. We found that using a variational formulation was essential to obtain smooth latent representations while training the encoder-decoder.

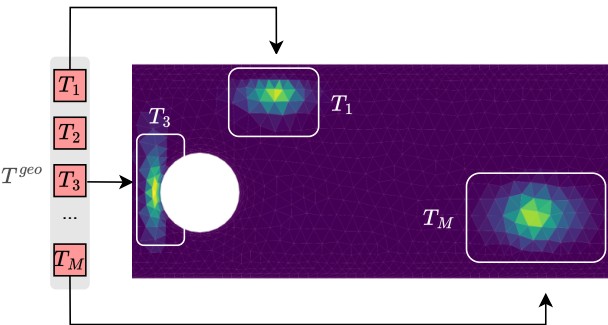

Figure 2: Spatial interpretation of the tokens through cross attention between $T^{geo}$ and $\boldsymbol{\gamma}(x)$ for each $x$ in the domain. Here we visualize the cross-attention of three different tokens for a given head. The cross attentions can have varying receptive fields depending on the geometries.

### 3.3 Transformer-based diffusion

Modeling the dynamics is performed in the latent $\boldsymbol{Z}$ space. This space encodes spatial information present in the original space while being a condensed, smaller-sized representation, allowing for reduced complexity dynamics modeling. As indicated, the dynamics can be unrolled auto-regressively in this space for any time horizon without the need to map back to the original space. We use absolute positional embeddings $E_{\text{pos}}$ and a linear layer to project onto a higher dimensional space: $\boldsymbol{Z}_{[0]} = \texttt{Linear}(\boldsymbol{Z}) + E_{\text{pos}}$. The backbone then applies several self-attention blocks, which process tokens as follows:

$$\boldsymbol{Z}_{[l+1]} \leftarrow \boldsymbol{Z}_{[l]} + \texttt{Attention}(\texttt{LayerNorm}(\boldsymbol{Z}_{[l]})) \tag{4}$$

$$\boldsymbol{Z}_{[l+1]} \leftarrow \boldsymbol{Z}_{[l+1]} + \texttt{FFN}(\texttt{LayerNorm}(\boldsymbol{Z}_{[l+1]})) \tag{5}$$

We found out that adding a diffusion component to the transformer helped enhance the stability and allowed longer forecasts. Diffusion steps are inserted between two time steps $t$ and $t + \Delta t$ of the time-marching process transformer. The diffusion steps are denoted by $k$ and are different from the ones of the time-marching process (several diffusion steps $k$ are performed between two time-marching steps $t$ and $t + \Delta t$).

We then use a conditional diffusion transformer architecture close to Peebles & Xie (2023) for $\mathcal{R}_\theta$, where we detail the main block in Appendix B. At diffusion step $k$, the input to the network is a sequence stacking the tokens at time $t$ and the current noisy targets estimate $(\boldsymbol{Z}^t, \tilde{\boldsymbol{Z}}_k^{t+\Delta t})$. See Appendix B, Figure 4 and Figure 5 for more details. To train the diffusion transformer $\mathcal{R}_\theta$, we freeze the encoder and decoder, and use the encoder to sample pairs of successive latent tokens $(\boldsymbol{Z}^t, \boldsymbol{Z}^{t+\Delta t})$. We employ the "v-predict" formulation of DDPM (Salimans & Ho, 2022) for training and sampling.

## 4 Experiments

In this section, we systematically evaluate the performance of our proposed model across various experimental settings, focusing on its ability to handle dynamics on both regular and irregular grids. First, we investigate the dynamics on regular grids, where we benchmark our model against state-of-the-art neural operators, including Fourier Neural Operators (FNO), ResNet, Neural Fields, and Transformers. This comparison highlights the efficacy of our approach in capturing complex spatio-temporal patterns on structured domains. Second, we extend our analysis to dynamics on irregular grids and shared geometries, emphasizing the model's extrapolation capabilities in data-constrained regimes. Here, we compare our results with Neural Fields and Transformers, demonstrating the robustness of our model in handling less structured and more complex spatial configurations. Lastly, we assess the model's capacity to process diverse geometries and underlying spatial representations by comparing its performance on irregular grids and different geometries. This evaluation highlights the flexibility and generalization ability of our model in encoding and learning from varied spatial domains, showcasing its potential in accurately representing and predicting dynamics across a wide range of geometric settings. We include additional results from ablation studies in Appendix C.6.

## 4.1 Dynamics on regular grids

We begin our analysis with dynamics modeling on regular grid settings. Though our model is targeted for complex geometries, we believe this scenario remains an important benchmark to assess the efficiency of surrogate models.

**Datasets** • **1D Burgers' Equation** (*Burgers*): Models shock waves, using a dataset with periodic initial conditions and forcing term as in Brandstetter et al. (2022). It includes 2048 training and 128 test trajectories, at resolutions of $(250, 100)$. We create sub-trajectories of 50 timestamps and treat them independently. • **2D Navier Stokes Equation**: for a viscous and incompressible fluid. We use the data from Li et al. (2021). The equation is expressed with the vorticity form on the unit torus: $\frac{\partial w}{\partial t} + u \cdot \nabla w = \nu \Delta w + f, \nabla u = 0$ for $x \in \Omega, t > 0$, where $\nu$ is the viscosity coefficient. We consider two different versions $\nu = 10^{-4}$ (*Navier-Stokes* $1 \times 10^{-4}$) and $\nu = 10^{-5}$ (*Navier-Stokes* $1 \times 10^{-5}$), and use train and test sets of 1000 and 200 trajectories with a base spatial resolution of size $64 \times 64$. We consider a horizon of $T = 30$ for $\nu = 10^{-4}$ and $T = 20$ for $\nu = 10^{-5}$ since the phenomenon is more turbulent. At test time, we use the vorticity at $t_0 = 10$ as the initial condition.

**Setting** We train all the models with supervision on the next state prediction to learn to approximate the time-stepping operator $u^{t+\Delta t} = \mathcal{G}(u^t)$. At test time, we unroll the dynamics auto-regressively with each model and evaluate the prediction with a relative $L_2$ error defined as $L_2^{\text{test}} = \frac{1}{N_{\text{test}}} \sum_{j \in \text{test}} \frac{||\hat{u}_j^{\text{trajectory}} - u_j^{\text{trajectory}}||_2}{||u_j^{\text{trajectory}}||_2}$.

**Baselines** We use a diverse panel of baselines including state of the art regular-grid methods such as FNO (Li et al., 2021) and ResNet (He et al., 2016; Lippe et al., 2023), flexible transformer architectures such as OFormer (Li et al., 2023), and GNOT (Hao et al., 2023), and finally neural-field based methods with DINO (Yin et al., 2022) and CORAL (Serrano et al., 2023).

**Results** Table 1 presents a comparison of model performance on the *Burgers*, *Navier-Stokes1e-4*, and *Navier-Stokes1e-5* datasets, with metrics reported in Relative $L_2$. Our method, AROMA, demonstrates excellent performance across the board, highlighting its ability to capture the dynamics of turbulent phenomena, as reflected in the *Navier-Stokes* datasets.

In contrast, DINO and CORAL, both global neural field models, perform poorly in capturing turbulent phenomena, exhibiting significantly higher errors compared to other models. This indicates their limitations in handling complex fluid dynamics. On the other hand, AROMA outperforms GNOT on all datasets, though it performs reasonably well compared to the neural field based method.

Table 1: **Model Performance Comparison** - Test results. Metrics in Relative $L_2$.

| Model | *Burgers* | *Navier-Stokes* $1 \times 10^{-4}$ | *Navier-Stokes* $1 \times 10^{-5}$ |
|---|---|---|---|
| FNO | $5.00 \times 10^{-2}$ | $\underline{1.53 \times 10^{-1}}$ | $\mathbf{1.24 \times 10^{-1}}$ |
| ResNet | $8.50 \times 10^{-2}$ | $3.77 \times 10^{-1}$ | $2.56 \times 10^{-1}$ |
| DINO | $4.57 \times 10^{-1}$ | $7.25 \times 10^{-1}$ | $3.72 \times 10^{-1}$ |
| CORAL | $6.20 \times 10^{-2}$ | $3.77 \times 10^{-1}$ | $3.11 \times 10^{-1}$ |
| GNOT | $1.28 \times 10^{-1}$ | $1.85 \times 10^{-1}$ | $1.65 \times 10^{-1}$ |
| OFormer | $\underline{4.92 \times 10^{-2}}$ | $1.36 \times 10^{-1}$ | $2.40 \times 10^{-1}$ |
| AROMA | $\mathbf{3.65 \times 10^{-2}}$ | $\mathbf{1.05 \times 10^{-1}}$ | $\mathbf{1.24 \times 10^{-1}}$ |

Regarding the regular-grid methods, ResNet shows suboptimal performance in the pure teacher forcing setting, rapidly accumulating errors over time during inference. FNO stands out as the best baseline, demonstrating competitive performance on all datasets. We hypothesize that FNO's robustness to error accumulation during the rollout can be attributed to its Fourier block, which effectively cuts off high-frequency components. Overall, the results underscore AROMA's effectiveness and highlight the challenges Neural Field-based models face in accurately modeling complex phenomena.

## 4.2 Dynamics on irregular grids with shared geometries

We continue our experimental analysis with dynamics on unstructured grids, where we observe trajectories only through sparse spatial observations over time. We adopt a data-constrained regime and show that our model can still be competitive with existing Neural Fields in this scenario.

**Datasets** To evaluate our framework, we utilize two fluid dynamics datasets commonly used as a benchmark for this task (Yin et al., 2022; Serrano et al., 2023) with unique initial conditions for each trajectory: • **2D Navier-Stokes Equation** (*Navier-Stokes* $1 \times 10^{-3}$): We use the same equation as

in Section 4.1 but with a higher viscosity coefficient $\nu = 1e-3$. We have 256 trajectories of size 40 for training and 32 for testing. We used a standard resolution of 64x64. • **3D Shallow-Water Equation** (*Shallow-Water*): This equation approximates fluid flow on the Earth's surface. The data includes the vorticity $w$ and height $h$ of the fluid. The training set comprises 64 trajectories of size 40, and the test set comprises 8 trajectories with 40 timestamps. We use a standard spatial resolution of $64 \times 128$.

**Setting** • **Temporal Extrapolation**: For both datasets, we split trajectories into two equal parts of 20 timestamps each. The first half is denoted as *In-t* and the second half as *Out-t*. The training set consists of *In-t*. During training, we supervise with the next state only. During testing, the model unrolls the dynamics from a new initial condition (IC) up to the end of *Out-t*, i.e. for 39 steps. Evaluation within the *In-t* horizon assesses the model's ability to forecast within the training regime. The *Out-t* evaluation tests the model's extrapolation capabilities beyond the training horizon. • **Sparse observations**: For the train and test set we randomly select $\pi$ percent of the available regular mesh to create a unique grid for each trajectory, both in the train and in the test. The grid is kept fixed along a given trajectory. While each grid is different, they maintain the same level of sparsity across trajectories. In our case, $\pi = 100\%$ amounts to the fully observable case, while in $\pi = 25\%$ each grid contains around 1020 points for *Navier-Stokes* $1 \times 10^{-3}$ and 2040 points for *Shallow-Water*.

**Baselines** We compare our model to OFormer (Li et al., 2023), GNOT (Hao et al., 2023), and choose DINO (Yin et al., 2022) and CORAL (Serrano et al., 2023) as the neural field baselines.

**Training and evaluation** During training, we only use the data from the training horizon (*In-t*). At test time, we evaluate the models to unroll the dynamics for new initial conditions in the training horizon (*In-t*) and for temporal extrapolation (*Out-t*).

**Results** Table 2 demonstrates that AROMA consistently achieves low MSE across all levels of observation sparsity and evaluation horizons for both datasets. Overall, our method performs best with some exceptions. On *Shallow-Water* our model is slightly outperformed by CORAL in the fully observed regime, potentially because of a lack of data. Similarly, on *Navier-Stokes* $1 \times 10^{-3}$ CORAL has slightly better scores in the very sparse regime $\pi = 5\%$. Overall, this is not surprising as meta-learning models excel in data-constrained regimes. We believe our geometry-encoding block is crucial for obtaining good representations of the observed values in the sparse regimes, potentially explaining the performance gap with GNOT and OFormer.

Table 2: **Temporal Extrapolation** - Test results. Metrics in MSE.

| $\mathcal{X}_{tr} \downarrow \mathcal{X}_{te}$ | dataset → | *Navier-Stokes* $1 \times 10^{-3}$ | | *Shallow-Water* | |
|---|---|---|---|---|---|
| | | *In-t* | *Out-t* | *In-t* | *Out-t* |
| | DINO | $2.51 \times 10^{-2}$ | $9.91 \times 10^{-2}$ | $4.15 \times 10^{-4}$ | $3.55 \times 10^{-3}$ |
| $\pi = 100\%$ | CORAL | $5.76 \times 10^{-4}$ | $3.00 \times 10^{-3}$ | $\mathbf{2.12 \times 10^{-5}}$ | $\mathbf{6.00 \times 10^{-4}}$ |
| | OFormer | $7.76 \times 10^{-3}$ | $6.39 \times 10^{-2}$ | $1.00 \times 10^{-2}$ | $2.23 \times 10^{-2}$ |
| | GNOT | $\underline{3.21 \times 10^{-4}}$ | $\underline{2.33 \times 10^{-3}}$ | $2.48 \times 10^{-4}$ | $2.17 \times 10^{-3}$ |
| | AROMA | $\mathbf{1.32 \times 10^{-4}}$ | $\mathbf{2.23 \times 10^{-3}}$ | $\underline{3.10 \times 10^{-5}}$ | $\underline{8.75 \times 10^{-4}}$ |
| | DINO | $3.27 \times 10^{-2}$ | $1.40 \times 10^{-1}$ | $4.12 \times 10^{-4}$ | $3.26 \times 10^{-3}$ |
| $\pi = 25\%$ | CORAL | $\underline{1.54 \times 10^{-3}}$ | $\underline{1.07 \times 10^{-2}}$ | $\underline{3.77 \times 10^{-4}}$ | $1.44 \times 10^{-3}$ |
| irregular grid | OFormer | $3.73 \times 10^{-2}$ | $1.60 \times 10^{-1}$ | $6.19 \times 10^{-3}$ | $1.40 \times 10^{-2}$ |
| | GNOT | $2.07 \times 10^{-2}$ | $6.24 \times 10^{-2}$ | $8.91 \times 10^{-4}$ | $\underline{4.66 \times 10^{-3}}$ |
| | AROMA | $\mathbf{7.02 \times 10^{-4}}$ | $\mathbf{6.31 \times 10^{-3}}$ | $\mathbf{1.49 \times 10^{-4}}$ | $\mathbf{1.02 \times 10^{-3}}$ |
| | DINO | $3.63 \times 10^{-2}$ | $1.35 \times 10^{-1}$ | $4.47 \times 10^{-3}$ | $9.88 \times 10^{-3}$ |
| $\pi = 5\%$ | CORAL | $\mathbf{2.87 \times 10^{-3}}$ | $\mathbf{1.48 \times 10^{-2}}$ | $\underline{2.72 \times 10^{-3}}$ | $\underline{6.58 \times 10^{-3}}$ |
| irregular grid | OFormer | $3.23 \times 10^{-2}$ | $1.12 \times 10^{-1}$ | $8.67 \times 10^{-3}$ | $1.72 \times 10^{-2}$ |
| | GNOT | $7.43 \times 10^{-2}$ | $1.89 \times 10^{-1}$ | $5.05 \times 10^{-3}$ | $1.49 \times 10^{-2}$ |
| | AROMA | $\underline{4.73 \times 10^{-3}}$ | $\underline{2.01 \times 10^{-2}}$ | $\mathbf{1.93 \times 10^{-3}}$ | $\mathbf{3.14 \times 10^{-3}}$ |

## 4.3 Dynamics on different geometries

Finally, we extend our analysis to learning dynamics over varying geometries.

**Datasets**  We evaluate our model on two problems involving non-convex domains, as described by Pfaff et al. (2021). Both scenarios involve fluid dynamics in a domain with an obstacle, where the area near the boundary conditions (BC) is more finely discretized. The boundary conditions are specified by the mesh, and the models are trained with various obstacles and tested on different, yet similar, obstacles. • **Cylinder** (*CylinderFlow*): This dataset simulates water flow around a cylinder using a fixed 2D Eulerian mesh, representing *incompressible* fluids. For each node $j$ in the mesh $\mathcal{X}$, we have data on the node position $x^{(j)}$, momentum $w(x^{(j)})$, and pressure $p(x^{(j)})$. Our task is to learn the mapping from $(w_t(x), p_t(x))_{x \in \mathcal{X}}$ to $(w_{t+\Delta t}(x), p_{t+\Delta t}(x))_{x \in \mathcal{X}}$ for a fixed $\Delta t$. • **Airfoil** (*AirfoilFlow*): This dataset simulates the aerodynamics around an airfoil, relevant for *compressible* fluids. In addition to the data available in the Cylinder dataset, we also have the fluid density $\rho(x^{(j)})$ for each node $j$. Our goal is to learn the mapping from $(w_t(x), p_t(x), \rho_t(x))_{x \in \mathcal{X}}$ to $(w_{t+\Delta t}(x), p_{t+\Delta t}(x), \rho_{t+\Delta t}(x))_{x \in \mathcal{X}}$. Each example in the dataset corresponds to a unique mesh. On average, there are 5233 nodes per mesh for *AirfoilFlow* and 1885 for *CylinderFlow*. We temporally subsample the original trajectories by taking one timestamp out of 10, forming trajectories of 60 timestamps. We use the first 40 timestamps for training (*In-t*) and keep the last 20 timestamps for evaluation (*Out-t*).

**Setting**  We train all the models with supervision on the next state prediction. At test time, we unroll the dynamics auto-regressively with each model and evaluate the prediction with a mean squared error (MSE) both in the training horizon *(In-t)* and beyond the training horizon *(Out-t)*.

**Results**  The results in Table 3 show that AROMA outperforms other models in predicting flow dynamics on both *CylinderFlow* and *AirfoilFlow* geometries, achieving the lowest MSE values across all tests. This indicates AROMA's superior ability to encode geometric features accurately. Additionally, AROMA maintains stability over extended prediction horizons, as evidenced by its consistently low *Out-t* MSE values.

Table 3: **Dynamics on different geometries** - Test results. MSE on normalized data.

| Model | CylinderFlow | | AirfoilFlow | |
|---|---|---|---|---|
| | *In-t* | *Out-t* | *In-t* | *Out-t* |
| CORAL | $4.458 \times 10^{-2}$ | $8.695 \times 10^{-2}$ | $1.690 \times 10^{-1}$ | $3.420 \times 10^{-1}$ |
| DINO | $1.349 \times 10^{-1}$ | $1.576 \times 10^{-1}$ | $3.770 \times 10^{-1}$ | $4.740 \times 10^{-1}$ |
| OFormer | $5.020 \times 10^{-1}$ | $1.080 \times 10^{0}$ | $5.620 \times 10^{-1}$ | $7.620 \times 10^{-1}$ |
| AROMA | $\mathbf{1.480 \times 10^{-2}}$ | $\mathbf{2.780 \times 10^{-2}}$ | $\mathbf{5.720 \times 10^{-2}}$ | $\mathbf{1.940 \times 10^{-1}}$ |

## 4.4   Long rollouts and uncertainty quantification

After training different models on *Burgers*, we compare them on long trajectory rollouts. We start from $t_0 = 50$ (i.e. use a numerical solver for 50 steps), and unroll our dynamics auto-regressively for 200 steps. Note that all the models were only trained to predict the next state. We plot the correlation over rollout steps of different methods, including our model without the diffusion process, in Figure 3. We can clearly see the gain in stability in using the diffusion for long rollouts. Still, the predictions will eventually become uncorrelated over time as the solver accumulates errors compared with the numerical solution. As we employ a generative model, we can generate several rollouts and estimate the uncertainty of the solver with standard deviations. We can see in Appendix Figure 11 that this uncertainty increases over time. This uncertainty is not a guarantee that the solution lies within the bounds, but is an indication that the model is not confident in its predictions.

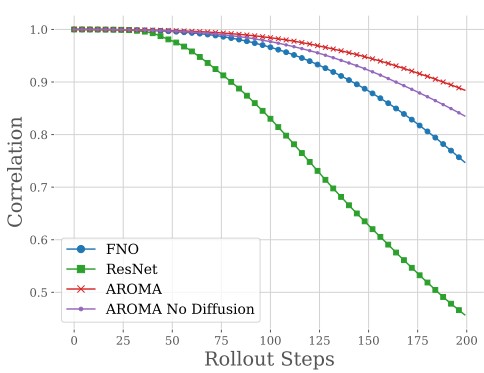

Figure 3: Correlation over time for long rollouts with different methods on *Burgers*

# 5 Related Work

Our model differs from existing models in the field of operator learning and more broadly from existing neural field architectures. The works most related to ours are the following.

**Neural Fields for PDE** Neural Fields have recently emerged as powerful tools to model dynamical systems. DINO (Yin et al., 2022) is a space-time continuous architecture based on a modulated multiplicative filter network (Fathony et al., 2021) and a NeuralODE (Chen & Zhang, 2019) for modeling the dynamics. DINO is capable of encoding and decoding physical states on irregular grids thanks to the spatial continuity of the INR and through auto-decoding (Park et al., 2019). CORAL is another neural-field based architecture, which tackles the broader scope of operator learning, also builds on meta-learning (Zintgraf et al., 2019; Dupont et al., 2022) to freely process irregular grids. CORAL and DINO are the most similar works to ours, as they are both auto-regressive and capable of processing irregular grids. On the other hand Chen et al. (2022) and Hagnberger et al. (2024) make use of spatio-temporal Neural Fields, for obtaining smooth and compact latent representations in the first or to directly predict trajectory solutions within a temporal horizon in the latter. Moreover, they either use a CNN or rely on patches for encoding the observations and are therefore not equipped for the type of tasks AROMA is designed for.

**Transformers for PDE** Several PDE solvers leverage transformers and cross-attention as a backbone for modeling PDEs. Transformers, which operate on token sequences, provide a natural solution for handling irregular meshes and point sets. Li et al. (2023) and Hao et al. (2023) introduced transformer architectures tailored for operator learning. Hao et al. (2023) incorporated an attention mechanism and employed a mixture of experts strategy to address multi-scale challenges. However, their architecture relies on linear attention without reducing spatial dimensions, resulting in linear complexity in sequence size, but quadratic in the hidden dimensions, which can be prohibitive for deep networks and large networks. Similarly, Li et al. (2023) utilized cross-attention to embed both regular and irregular meshes into a latent space and applied a recurrent network for time-marching in this latent space. Nonetheless, like GNOT, their method operates point-wise on the latent space. Transolver (Wu et al., 2024) decomposes a discrete input function into a mixture of "slices," each corresponding to a prototype in a mixture model, with attention operating in this latent space. This approach, akin to our model, reduces complexity. However, it has not been designed for temporal problems. (Alkin et al., 2024) recently proposed a versatile model capable of operating on Eulerian and Lagrangian (particles) representations. They reduce input dimensionality by aggregating information from input values onto "supernodes" selected from the input mesh via message passing while decoding is performed with a Perceiver-like architecture. In contrast, AROMA performs implicit spatial encoding with cross-attention to encode the geometry and aggregate obsevation values. Finally, their training involves complex end-to-end optimization, whereas we favor two simple training steps that are easier to implement.

# 6 Conclusion and Limitations

AROMA offers a novel and flexible neural operator approach for modeling the spatio-temporal evolution of physical processes. It is able to deal with general geometries and to forecast at any position of the spatial domain. It incorporates in an encode-process-decode framework attention mechanisms, a latent diffusion transformer for spatio-temporal dynamics and neural fields for decoding. Thanks to a very compact spatial encoding, its complexity is lower than most SOTA models. Experiments with small-size datasets demonstrate its effectiveness. Its reduced complexity holds potential for effective scaling to larger datasets. As for the limitations, the performance of AROMA are still to be demonstrated on larger and real world examples. Moreover, like all dynamical models that operate over a latent space, the reconstruction capabilities of the decoder is a bottleneck for the rollout accuracy. Since the encoder and decoder are learning spatial relationships from scratch, conferring the framework a high flexibility, the training efficiency does not match that of CNN-based auto-encoders on regular grids. We therefore believe there could be further improvements to be made to achieve a similar performance while keeping the same level of flexibility. Finally, even though our model has some potential for uncertainty modeling, this aspect has still to be further explored and analyzed.

## Acknowledgements

We acknowledge the financial support provided by DL4CLIM (ANR-19-CHIA-0018-01), DEEP-NUM (ANR-21-CE23-0017-02), PHLUSIM (ANR-23-CE23-0025-02), and PEPR Sharp (ANR-23-PEIA-0008, ANR, FRANCE 2030). This work was granted access to the HPC resources of IDRIS under the allocations 2023-AD011013522R1, 2023-AD011013332R1, 2023-AD011015133R1, 2023-A0161015133 made by GENCI.

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

## A  Extended Related Work

**Diffusion models for PDE**  Recently, diffusion models have experienced significant growth and success in generative tasks, such as image or video generation (Ho et al., 2020). This success has motivated their application to physics prediction. Rühling Cachay et al. (2023) propose DYffusion, a framework that adapts the diffusion process to spatio-temporal data for forecasting on long-time roll-outs, by performing diffusion-like timesteps in the physical time dimension. PDE-Refiner (Lippe et al., 2023) is a CNN-based method that uses diffusion to stabilize prediction rollouts over long trajectories. Compared to these methods, we perform diffusion in a latent space, reducing the computational cost; and leverage the advanced modeling capabilities of transformers.

**Local Neural Fields**  We are not the first work that proposes to leverage locality to improve the design of neural fields. In a different approach, Bauer et al. (2023) proposed a grid-based latent space where the modulation function $\phi$ is dependent on the query coordinate $x$. This concept enables the application of architectures with spatial inductive biases for generation on the latent representations, such as a U-Net Denoiser for diffusion processes. Similarly, Lee et al. (2023) developed a locality-aware, generalizable Implicit Neural Representation (INR) with demonstrated capabilities in generative modeling. Both of these architectures assume regular input structures, be it through patching methods or grid-based layouts.

## B  Implementation details

**Diffusion transformer**  We illustrate how our diffusion transformer is trained and used at inference in Figure 4 and Figure 5. We provide the diffusion step $k$ which acts as a conditioning input for the diffusion model. We use an exponential decrease for the noise level as in Lippe et al. (2023) i.e. $\alpha_k = 1 - \sigma_{\min}^{k/K}$. We use the same diffusion transformer block as in Peebles & Xie (2023), which relies on amplitude and shift modulations from the diffusion timestamp $k$:

$$\alpha^{(1)}, \beta^{(1)}, \gamma^{(1)} \leftarrow \mathrm{MLP}_1(k) \tag{6}$$

$$\alpha^{(2)}, \beta^{(2)}, \gamma^{(2)} \leftarrow \mathrm{MLP}_2(k) \tag{7}$$

$$\boldsymbol{Z}_{[l+1]} \leftarrow \boldsymbol{Z}_{[l]} + \alpha^{(1)} \cdot \texttt{Attention}(\gamma^{(1)} \cdot \texttt{LayerNorm}(\boldsymbol{Z}_{[l]}) + \beta^{(1)}) \tag{8}$$

$$\boldsymbol{Z}_{[l+1]} \leftarrow \boldsymbol{Z}_{[l+1]} + \alpha^{(2)} \cdot \texttt{FFN}(\gamma^{(2)} \cdot \texttt{LayerNorm}(\boldsymbol{Z}_{[l+1]} + \beta^{(2)})) \tag{9}$$

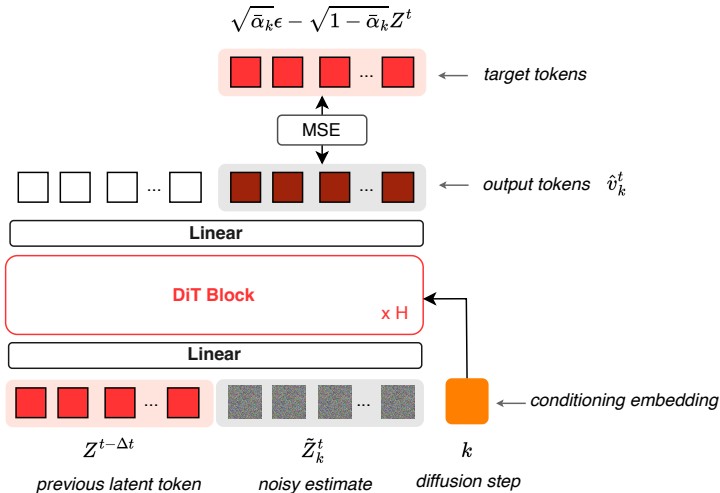

Figure 4: During training, we noise the next-step latent tokens $\boldsymbol{Z}^{t+\Delta t}$ and train the transformer to predict the "velocity" of the noise. Each DIT block is implemented as in Peebles & Xie (2023).

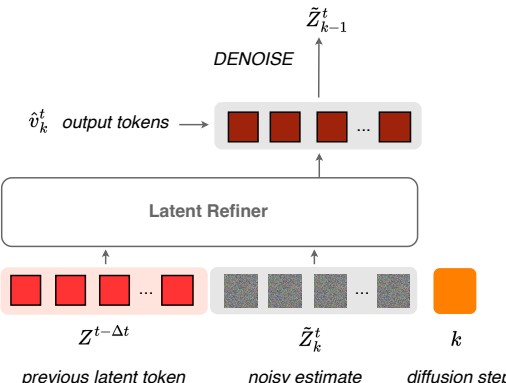

Figure 5: At inference, we start from $\tilde{\boldsymbol{Z}}_K^{t+\Delta t} \sim \mathcal{N}(0, I)$ and reverse the diffusion process to denoise our prediction. We set our prediction $\hat{\boldsymbol{Z}}^{t+\Delta t} = \tilde{\boldsymbol{Z}}_0^{t+\Delta t}$.

**Encoder-Decoder**    We provide a more detailed description of the encoder-decoder pipeline in Figure 6.

**Local INR**    We show the implementation of our local INR, both with single-band frequency and multi-band frequency, in Figure 7 and Figure 8. The cross-attention mechanism enables to retrieve a local feature vector $\mathbf{f}_q(x)$ for each query position $x$. We then use an MLP to decode this feature vector to retrieve the output value. In practice, we retrieve several feature vectors corresponding each to separate frequency bandwidths. In this case, we concatenate the feature vectors before decoding them with the MLP.

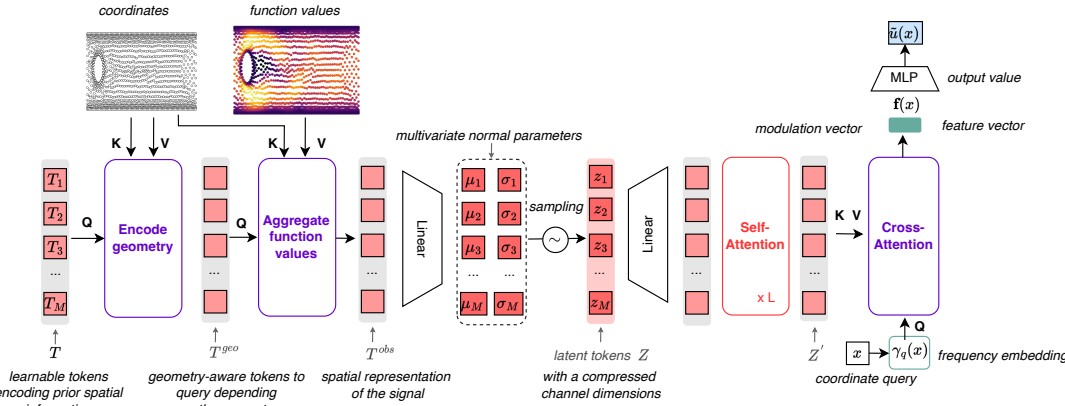

Figure 6: Architecture of our encoder and decoder. We regularize the architecture as a variational auto-encoder. Cross-attention layers are used to aggregate the $N$ observations into $M$ latent tokens, and to expand the $M$ processed tokens to the queried values. We use a bottleneck layer to reduce the channel dimension of the latent space.

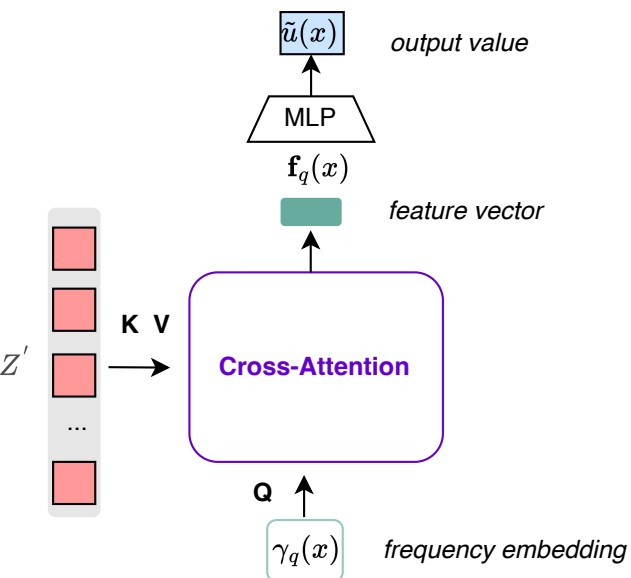

Figure 7: Single-band local INR decoder

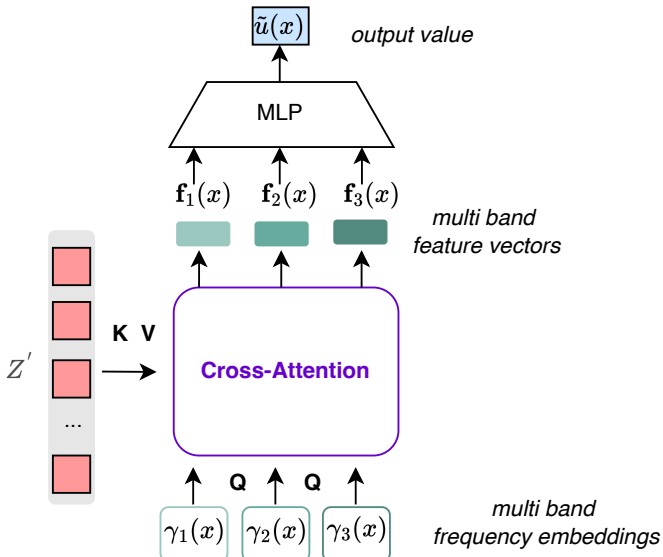

Figure 8: Multi-band local INR decoder

## B.1 Hyperparameters

We detail the values of the hyperparameters used on each dataset: Table 5 presents the hyperparameters of the Encoder-Decoder, while Table 4 presents the hyperparameters of the Diffusion Transformer. We use a cosine scheduler for the tuning learning rate for both trainings, with an initial maximum learning rate of $10^{-3}$ annealing to $10^{-5}$. All experiments were performed with an NVIDIA TITAN RTX.

For the diffusion transformer, we use $K = 3$ diffusion steps for all experiments and only vary the minimum noise $\sigma_{\min}$.

Table 4: Diffusion Transformer Hyperparameters for Different Datasets

| Hyperparameters | Burgers | NS1e-3 | NS1e-4 | NS1e-5 | Shallow-water | Cylinder-Flow | Airfoil-Flow |
|---|---|---|---|---|---|---|---|
| hidden_size | 128 | 128 | 128 | 128 | 128 | 128 | 128 |
| depth | 4 | 4 | 4 | 4 | 4 | 4 | 4 |
| num_heads | 4 | 4 | 4 | 4 | 4 | 4 | 4 |
| mlp_ratio | 4.0 | 4.0 | 4.0 | 4.0 | 4.0 | 4.0 | 4.0 |
| min_noise | 1e-2 | 1e-2 | 1e-3 | 1e-3 | 1e-3 | 1e-3 | 1e-3 |
| denoising_steps | 3 | 3 | 3 | 3 | 3 | 3 | 3 |
| epochs | 2000 | 2000 | 2000 | 2000 | 2000 | 2000 | 2000 |

For the encoder-decoder, we have the following hyperparameters:

- hidden_dim: The number $h$ of neurons at each hidden layer.

- num_self_attentions: The number of Self Attention layers used for the decoder.

- num_latents: The number $M$ of latent tokens used to spatially project the objervations and geometries.

- latent_dim: The dimension $c$ of each latent token.

- latent_heads: The number of heads use for the Self Attention layers.

- latent_dim_head: The dimension of each head in a Self Attention layer.

- cross_heads: The number of heads use for the Cross Attention layers.

- cross_dim_head: The dimension of each head in a Cross Attention layer.

- dim: The number of neurons used in the MLP decoder.

- depth_inr: The number of layers in the MLP decoder.

- frequencies: The different frequencies used for the local INR. We use base 2 for all experiments and select 16 frequencies in logarithmic scale per level. For example, [3, 4, 5] means that we construct 3 frequency embedding vectors, the first $\gamma_1(x) = (\cos(2^0\pi x), \sin(2^0\pi x), \ldots, \cos(2^3\pi x), \sin(2^3\pi x))$, for the second $\gamma_2 = (\cos(2^3\pi x), \sin(2^3\pi x), \ldots, \cos(2^4\pi x), \sin(2^4\pi x))$, and for the third $\gamma_3 = (\cos(2^4\pi x), \sin(2^4\pi x), \ldots, \cos(2^5\pi x), \sin(2^5\pi x))$

- dropout_sequence: The ratio of points that are ignored by the encoder.

- feature_dim: The dimension of the feature vector.

- encode_geo: If we use a cross-attention block to encode the geometry.

- max_encoding_freq: The maximum frequency used for the frequency embedding $\gamma$ of the encoder.

- kl_weight: The weight $\beta$ used for the VAE training.

- epochs: Number of training epochs.

The most important hyperparameter of the encoder-decoder is the number of tokens $M$ that are used to aggregate the observations and geometries. We show the impact it has on the quality of reconstructions in Table 6.

Table 5: Hyperparameters of the Encoder-Decoder for Different Datasets

| Hyperparameters | Burgers | NS1e-3 | NS1e-4 | NS1e-5 | Shallow-water | Cylinder-Flow | Airfoil-Flow |
|---|---|---|---|---|---|---|---|
| hidden_dim | 128 | 128 | 128 | 128 | 128 | 128 | 128 |
| num_self_attentions | 2 | 2 | 2 | 3 | 2 | 2 | 3 |
| num_latents | 32 | 32 | 256 | 256 | 32 | 64 | 64 |
| latent_dim | 8 | 16 | 16 | 16 | 16 | 16 | 16 |
| latent_heads | 4 | 4 | 4 | 4 | 4 | 4 | 4 |
| latent_dim_head | 32 | 32 | 32 | 32 | 32 | 32 | 32 |
| cross_heads | 4 | 4 | 4 | 4 | 4 | 4 | 4 |
| cross_dim_head | 32 | 32 | 32 | 32 | 32 | 32 | 32 |
| dim | 128 | 128 | 128 | 128 | 64 | 128 | 128 |
| depth_inr | 3 | 3 | 3 | 3 | 3 | 3 | 3 |
| frequencies | [3, 4, 5] | [2, 3] | [3, 4, 5] | [3, 4, 5] | [2, 3] | [3, 4, 5] | [3, 4, 5] |
| dropout_sequence | 0.1 | 0.1 | 0.1 | 0.1 | 0.1 | 0.1 | 0.1 |
| feature_dim | 16 | 16 | 16 | 16 | 16 | 16 | 16 |
| encode_geo | False | True | False | False | True | True | True |
| max_encoding_freq | 4 | 4 | 4 | 4 | 5 | 4 | 5 |
| kl_weight | 1e-4 | 1e-4 | 1e-4 | 1e-5 | 1e-5 | 1e-5 | 1e-5 |
| epochs | 5000 | 5000 | 5000 | 5000 | 5000 | 5000 | 5000 |

# C  Additional results

## C.1  Time complexity analysis

We denote $N$ as the number of observations of $\boldsymbol{u}$, $M$ as the number of tokens used to compress the information, $T$ as the number of autoregressive calls in the rollout, $K$ as the number of refinement steps, and $d$ as the number of channels used in the attention mechanism. The most computationally expensive operations in our architecture are the cross-attention and self-attention blocks. For simplification, we omit the geometry encoding block in this study.

The cost of the cross-attention in the encoder is $O(NMd)$, and similarly, the cost of the cross-attention in the decoder is $O(NMd)$. Let $L_1$ and $L_2$ represent the number of layers in the decoder and diffusion transformer, respectively. The cost of the self-attention layers in the decoder is $O(L_1M^2d)$, while in the diffusion transformer, it is $O(4L_2M^2d)$.

To unroll the dynamics, we encode the initial condition, obtain the predictions in the latent space, and then decode in parallel, yielding a total cost of $O((2N + 4KTL_2M + L_1M)Md)$. As expected, our architecture has linear complexity in the number of observations through the cross-attention layers. In contrast, GNOT relies on linear attention, resulting in a time complexity of $O((LN)d^2)$ for each prediction, where $L$ is the depth of the network. At inference, the cost per step along a trajectory is $LNd^2$ for GNOT, compared to $4KL_2M^2d$ for AROMA.

For instance, using $K = 3$, $M = 64$, $N = 4096$, and $d = 128$, GNOT's cost is approximately 10 times that of AROMA for each prediction throughout the rollout. Therefore AROMA is more efficient when $M \ll N$.

## C.2  Encoding interpretation

We provide in Figure 9 a qualitative analysis through cross-attention visualizations how the geometry encoding block helps to capture the geometry of the domain. In the first cross-attention block, the query tokens $\mathbf{T}$ are not aware of the geometry and therefore attend to large regions of the domains. This lets the model understand, where the boundaries of the domain are and therefore where the cylinder is. Once the query tokens have aggregated the mesh information, the cross attention between $\mathbf{T}^{\text{geo}}$ and the positions are sharper and depend on the geometry.

## C.3  Example rollouts

We show examples of rollout predictions using AROMA on *Burgers* dataset in Figure 10, on *Navier-Stokes* $1 \times 10^{-3}$ dataset in Figure 12 and on *CylinderFlow* in Figure 13. AROMA returns predictions that remain stable and accurate, even outside the training time horizon.

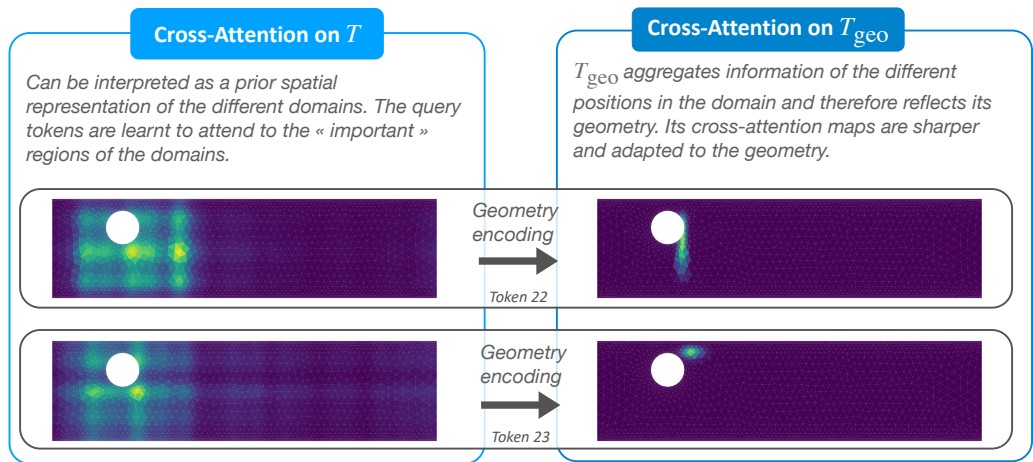

Figure 9: Evolution of the cross-attention maps between the geometry encoding stage and the observation encoding stage. Blue means the cross-attention value is close to zero while yellow means the cross-attention score is close to one.

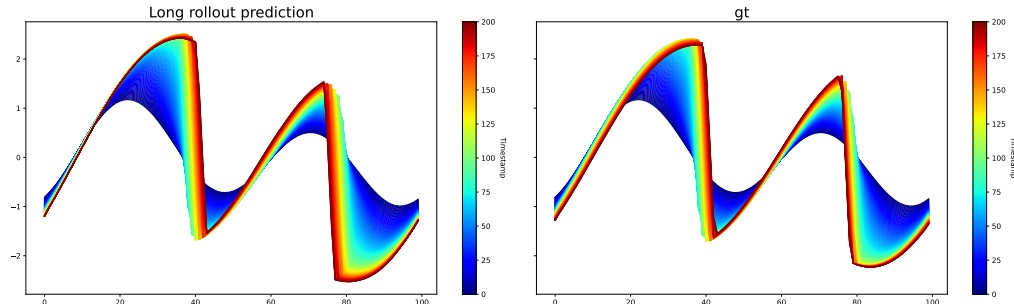

Figure 10: Test example long rollout trajectory with AROMA on *Burgers*. Left is the predicted trajectory and right is the ground truth.

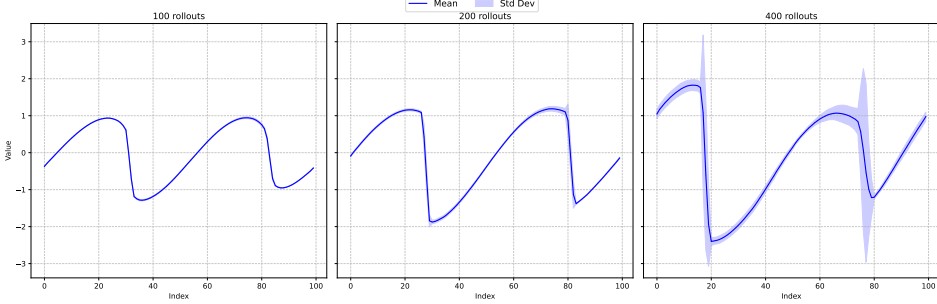

Figure 11: Uncertainty of AROMA over rollout steps. The blue line is the mean prediction while the blue shade represents the mean $\pm\, 3 \times$ standard deviation.

For *Navier-Stokes*, we show an example of test trajectory in the training horizon (Figure 12a) and in extrapolation (Figure 12b).

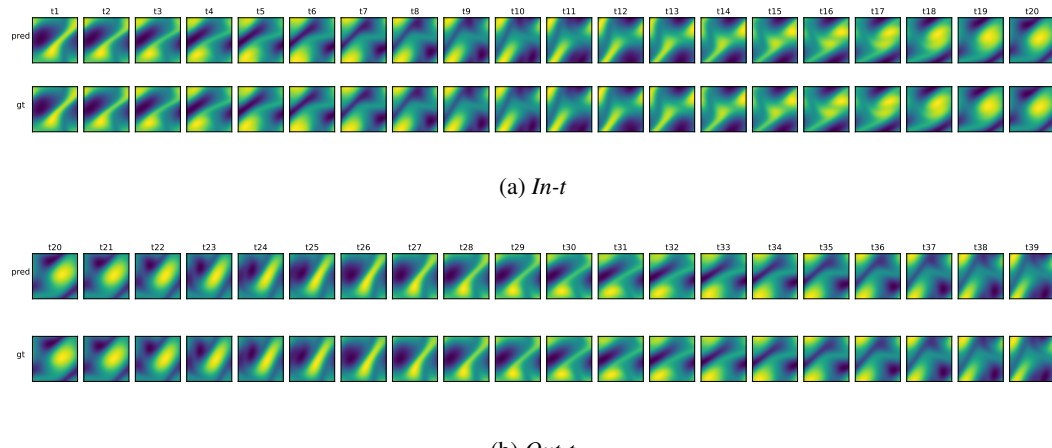

(a) *In-t*

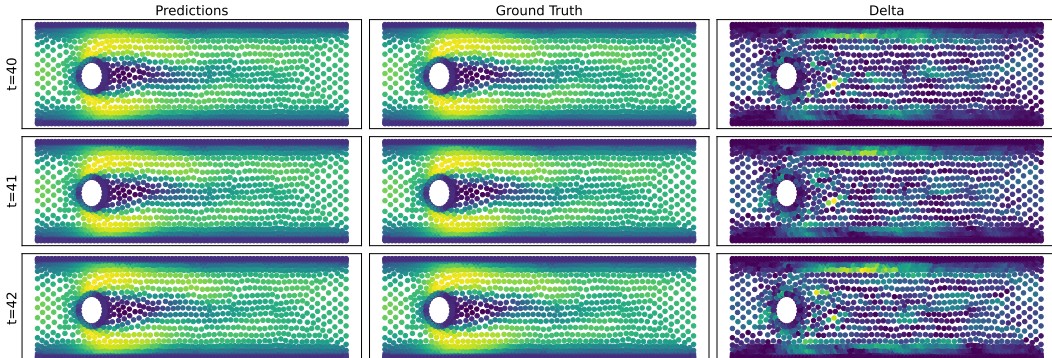

(b) *Out-t*

Figure 12: Test example rollout trajectories with AROMA on *Navier-Stokes* $1 \times 10^{-3}$. **Top:** predicted trajectory on *In-t*. **Bottom:** trajectory on *Out-t*. First row in each subfigure shows the prediction, the second row shows the ground truth.

Figure 13: Visualization of AROMA's predictions on *Cylinder* for (*Out-t*). The left panel shows the prediction, the middle panel displays the ground truth, and the right panel is the point-wise error.

## C.4 Scaling experiments

In Figure 14, we compare the reconstruction and prediction capabilities of CORAL and AROMA on *Navier-Stokes* $1 \times 10^{-4}$ given the number of training trajectories. As evidenced, our architecture outperforms CORAL significantly when the number of trajectories is greater than $10^3$, highlighting its efficacy in handling large amounts of data..

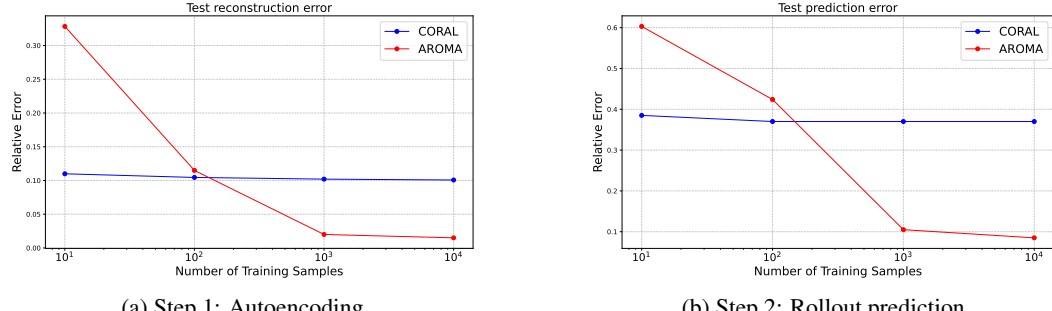

(a) Step 1: Autoencoding                    (b) Step 2: Rollout prediction

Figure 14: Scaling comparison of AROMA & CORAL: relative $L_2$ error with respect to the number of training trajectories

## C.5 Spatial tokens perturbation analysis

To validate the spatial interpretation of our latent tokens, we establish a baseline code $\boldsymbol{Z}^0$, and introduce perturbations by sequentially replacing the $j$-th token, $\boldsymbol{z}_j^0$, with subsequent tokens along the trajectory, denoted as $\boldsymbol{z}_j^1, \boldsymbol{z}_j^2, \ldots, \boldsymbol{z}_j^t$. Thus, the perturbed tokens mirror $\boldsymbol{Z}^0$ in all aspects except for the $j$-th token, which evolves according to the true token dynamics. We show reconstruction visualizations of the perturbed tokens in figs. 15 to 22. On the right side, we show the groundtruth of the trajectory. On the left side, is the change in AROMA's prediction in response to the token perturbation. These figures show that the perturbation of a token only impacts the reconstructed field locally, which validates the spatial structure of our tokens. Additionally, we can notice some interesting effects of the token perturbations near the boundaries in figs. 17 and 22: our encoder-decoder has discovered from data and without explicit supervision that the solutions had periodic boundary conditions by leveraging the encoded geometry and the function values. This validates the architecture of our cross-attention module between the function values, the spatial coordinates and the geometry-aware tokens.

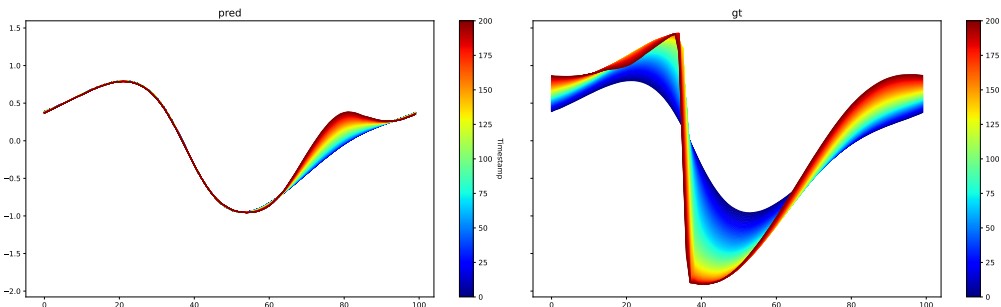

Figure 15: Perturbation analysis on *Burgers*. Token 0.

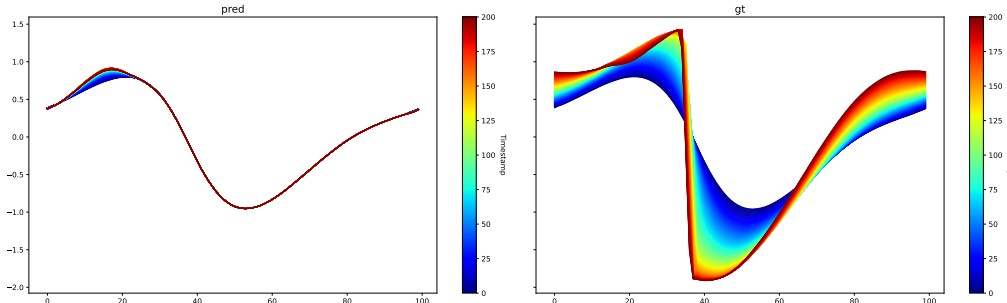

Figure 16: Perturbation analysis on *Burgers*. Token 1.

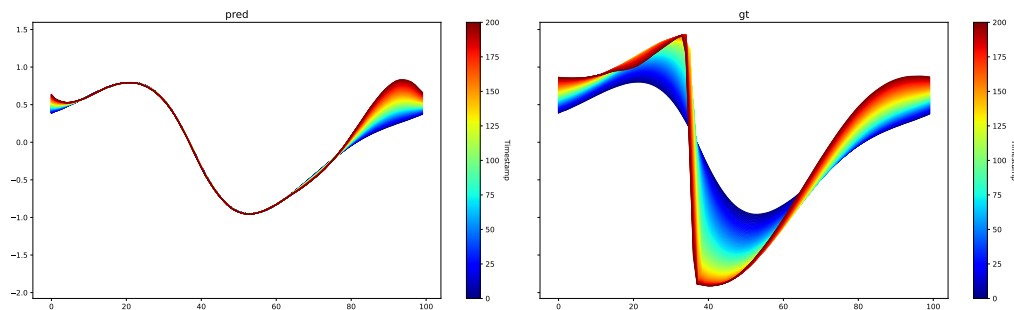

Figure 17: Perturbation analysis on *Burgers*. Token 2.

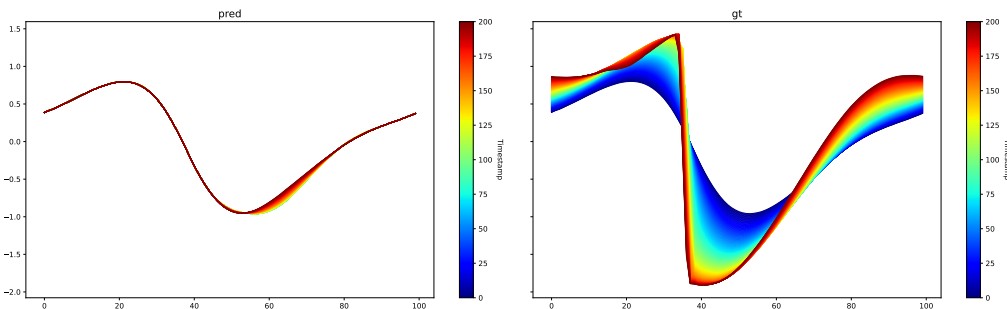

Figure 18: Perturbation analysis on *Burgers*. Token 3.

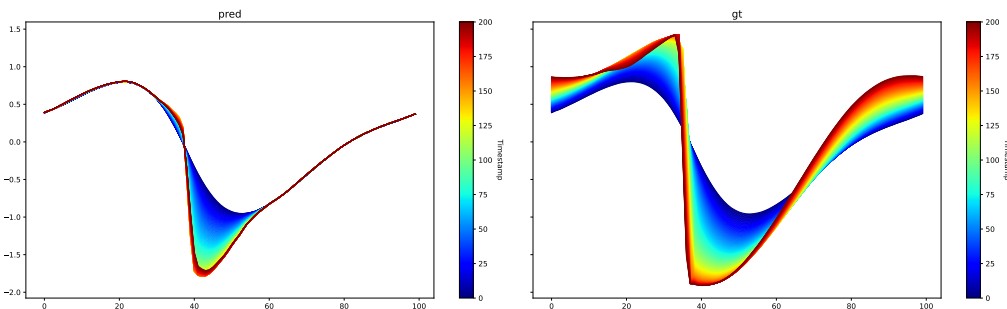

Figure 19: Perturbation analysis on *Burgers*. Token 5.

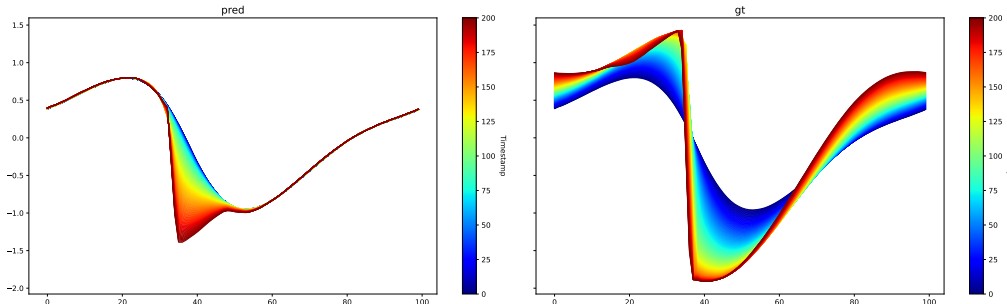

Figure 20: Perturbation analysis on *Burgers*. Token 6.

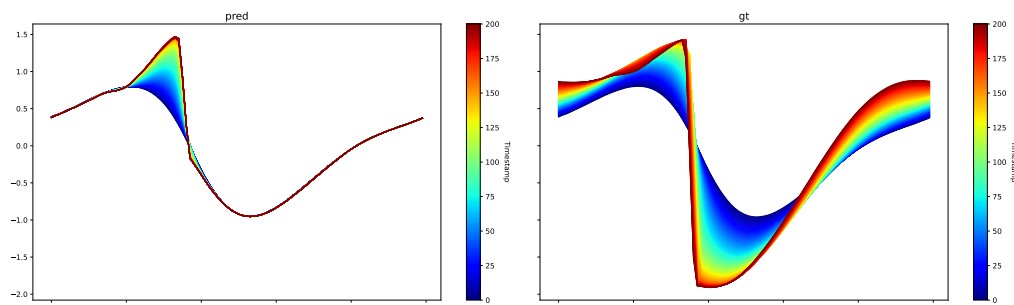

Figure 21: Perturbation analysis on *Burgers*. Token 7.

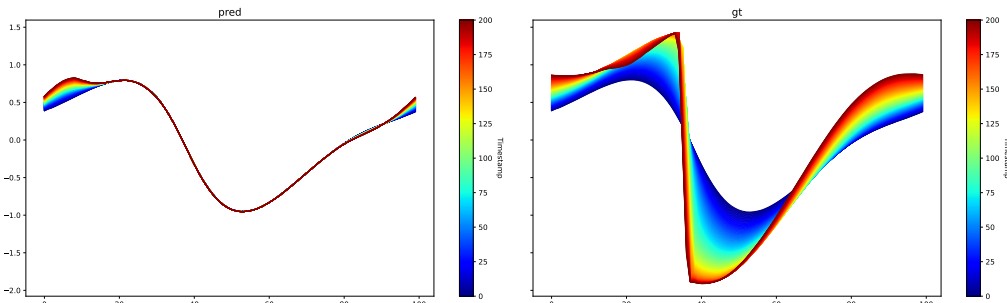

Figure 22: Perturbation analysis on *Burgers*. Token 8.

## C.6 Ablation studies

**Number of tokens** $M$   We show the impact of the number of latent tokens on the *Navier-Stokes* $1 \times 10^{-4}$ dataset in Table 6. We train our auto-encoder with 10000 trajectories. We can see that the performance increases with the number of tokens.

| #Latent Tokens | Test Reconstruction error |
|---|---|
| 64 | 0.02664 |
| 128 | 0.0123 |
| 256 | 0.01049 |

Table 6: Influence of the number of latent tokens on the test reconstruction capabilities on *Navier-Stokes* $1 \times 10^{-4}$. Performance in Relative $L_2$ Error.

**Auto-encoding vs VAE**   Our framework can also be used without the KL regularization, and could potentially be employed with other forms of regularization, such as L2 regularizaton or vector-quantization (Oord et al., 2017). We investigated in Table 7 the impact the KL regularization had on the overall rollout performance, and selected an autoencoder with L2 regularization (weight decay) as baseline. Our conclusion is that using an autoencoder with L2 regularization is a viable alternative to the VAE in some cases for achieving a smooth latent space. The autoencoder demonstrated superior performance on two datasets (*Burgers* and *Navier-Stokes* $1 \times 10^{-4}$, explained by its lower reconstruction errors, which translate into better rollout performance. However, for the more challenging *Navier-Stokes* $1 \times 10^{-5}$ case, the autoencoder's latent space exhibited high variance, which may explain the observed performance difference with the VAE.

**No-diffusion vs diffusion**   As an ablation, we also measured the influence of the diffusion formulation on the rollout accuracy by comparing to the same transformer architecture trained directly with an MSE on the mean tokens. The deterministic version of AROMA shows consistently robust performance and even surpasses the diffusion version on the *Navier-Stokes* $1 \times 10^{-4}$ case (Table 7). This demonstrates that the latent tokens obtained with AROMA contain meaningful information for dynamics modeling. On the other hand, the deterministic version yields less accurate long rollouts on *Burgers* or *KS* in Figure 23 and Figure 3. Note that using diffusion allows us to model the trajectory distribution, which opens the way to infer statistics on this distribution. This is key, for example, when modeling uncertainty, which is a critical problem for these models.

**Latent MLP vs Latent Transformer**   Modeling interactions at the local and global levels is key to learn the dynamics faithfully. Experiments using MLPs (Table 7) as time steppers which do not consider interactions between tokens lead to significantly lower performance compared to transformers.

Table 7: **Ablation Study**. Metrics in Relative $L_2$ on the test set.

| Model | Burgers | Navier-Stokes $1 \times 10^{-4}$ | Navier-Stokes $1 \times 10^{-5}$ |
|---|---|---|---|
| AROMA + auto-encoding | $\mathbf{3.43 \times 10^{-2}}$ | $\mathbf{5.02 \times 10^{-2}}$ | $2.10 \times 10^{-1}$ |
| AROMA w/o diffusion | $4.31 \times 10^{-2}$ | $7.50 \times 10^{-2}$ | $1.28 \times 10^{-1}$ |
| AROMA + mlp | $1.11 \times 10^{-1}$ | $1.00 \times 10^{0}$ | $8.25 \times 10^{-1}$ |
| AROMA | $3.65 \times 10^{-2}$ | $1.05 \times 10^{-1}$ | $\mathbf{1.24 \times 10^{-1}}$ |

## C.7 Kuramoto-Sivashinsky : a failure case

We conducted additional experiments on a chaotic 1D PDE, the Kuramoto-Sivashinsky (*KS*) equation. We found that AROMA currently struggles with dynamics that exhibit chaotic phenomena and non-decaying spectra, as shown in Figure 23. The primary limitation appears to be the reconstruction capabilities of the encoder-decoder. For the *KS* equation, we found that obtaining reconstructions with an MSE in the range of 1e-10 to 1e-12 was necessary for accurate spectrum reconstruction. Like all models leveraging a reduced latent representation space, AROMA inherently loses some of

the fine-grained details necessary for accurately capturing chaotic behavior. The diffusion framework slightly improves the high correlation time compared to the deterministic version, however the main bottleneck comes from the decoder (Table 8). In conclusion, while AROMA performs very well on simpler dynamics, dealing with chaotic phenomena requires more involved modeling that explicitly targets the chaotic component. Note that using dedicated modules for this purpose is a current practice in fluid dynamics - e.g. LES (Large eddy simulation).

Table 8: Test results on the **KS equation**. The evaluated metrics include: 1-step prediction MSE, MSE over the entire rollout (160 timestamps), and the duration for which the correlation between the generated samples and the ground truth remains above 0.8.

| Baseline | 1 step. MSE | Rollout MSE | Corr. $\geq 0.8$ |
|---|---|---|---|
| AROMA w/o diffusion | $\mathbf{1.25 \times 10^{-5}}$ | $2.20 \times 10^0$ | 32.8s |
| AROMA | $1.81 \times 10^{-5}$ | $\mathbf{2.07 \times 10^0}$ | **36.0s** |

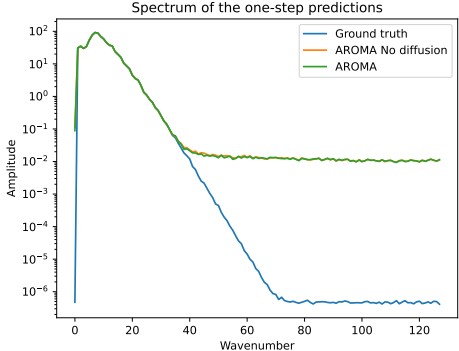 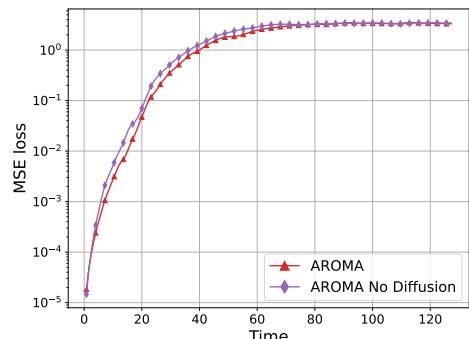

(a) Spectrum of AROMA's 1-step predictions vs ground truth.

(b) Comparison of the MSE loss ($\downarrow$) with and without diffusion.

Figure 23: Qualitative results on **KS equation**.

## C.8 Latent space dynamics

For *Navier-Stokes*, we show how the mean (Figure 24) and standard deviation tokens (Figure 25) evolve over time for a given test trajectory. We show the predicted trajectory of the latent tokens $Z$ in the latent space in Figure 26. In practice, the tokens where the logvar is 0 (i.e. a high variance) on Figure 25 do not impact the prediction (Rolinek et al., 2019). We can therefore see, that ouf of the 16 tokens used, the most influential ones are Token 6, 7, 8, 9, 15, 16, as they clearly exhibit non-noisy patterns.

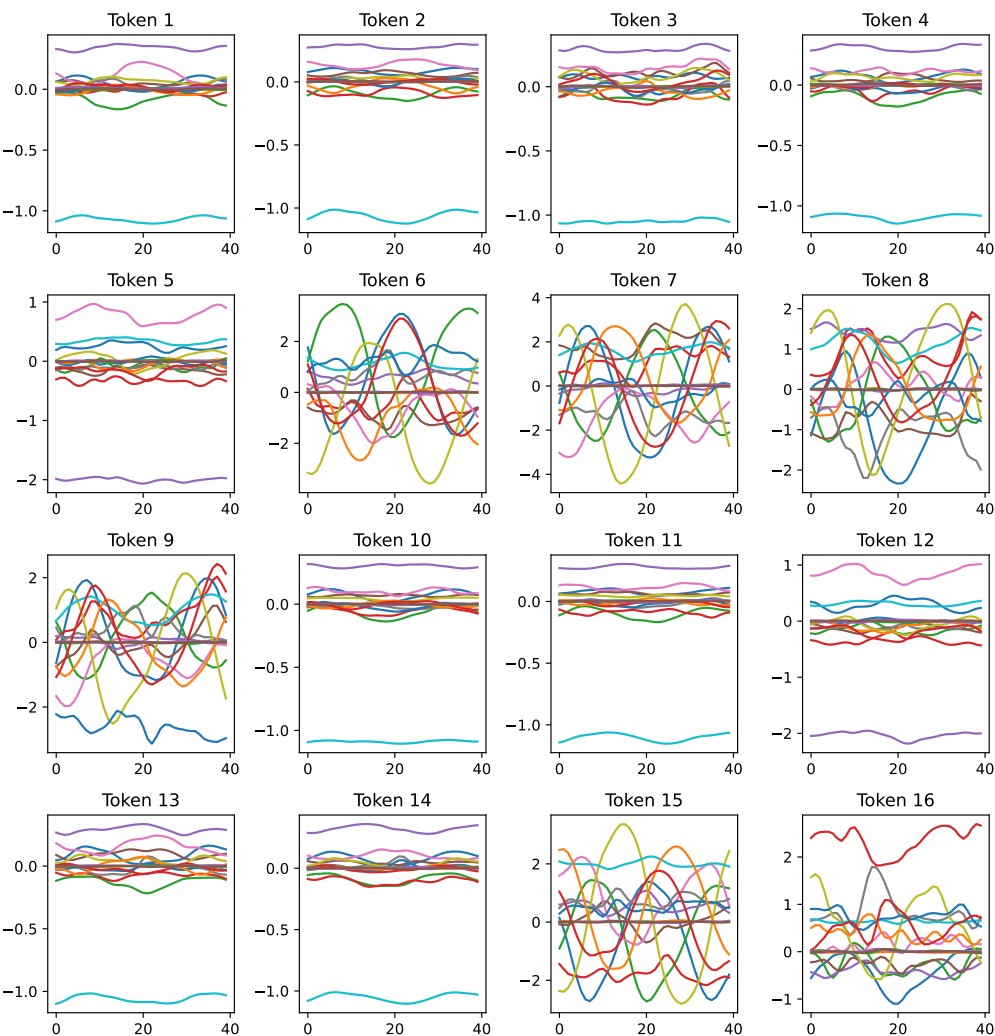

Figure 24: Latent space dynamics on *Navier-Stokes 1e-3* - Mean tokens over time. Each color line is a different token channel.

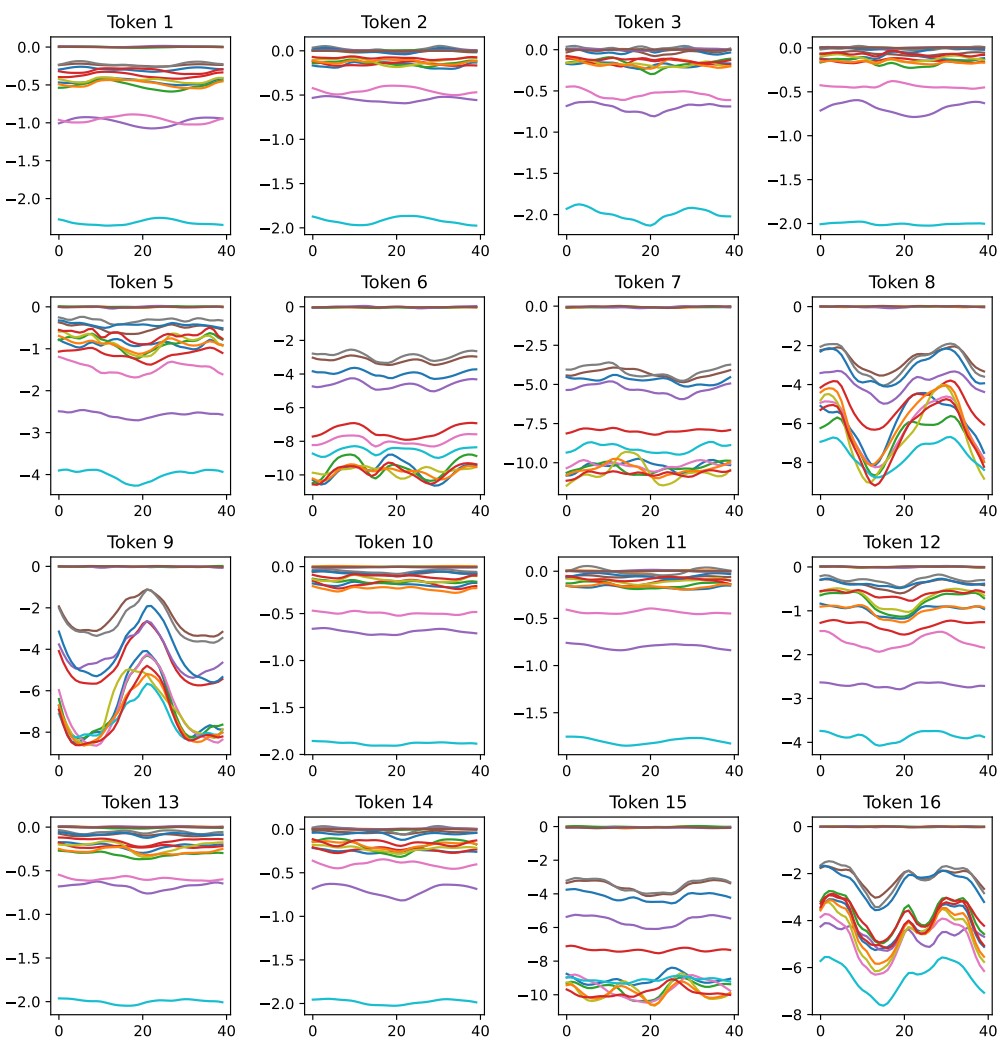

Figure 25: Latent space dynamics on *Navier-Stokes* - Logvar tokens over time. Each color line is a different token channel.

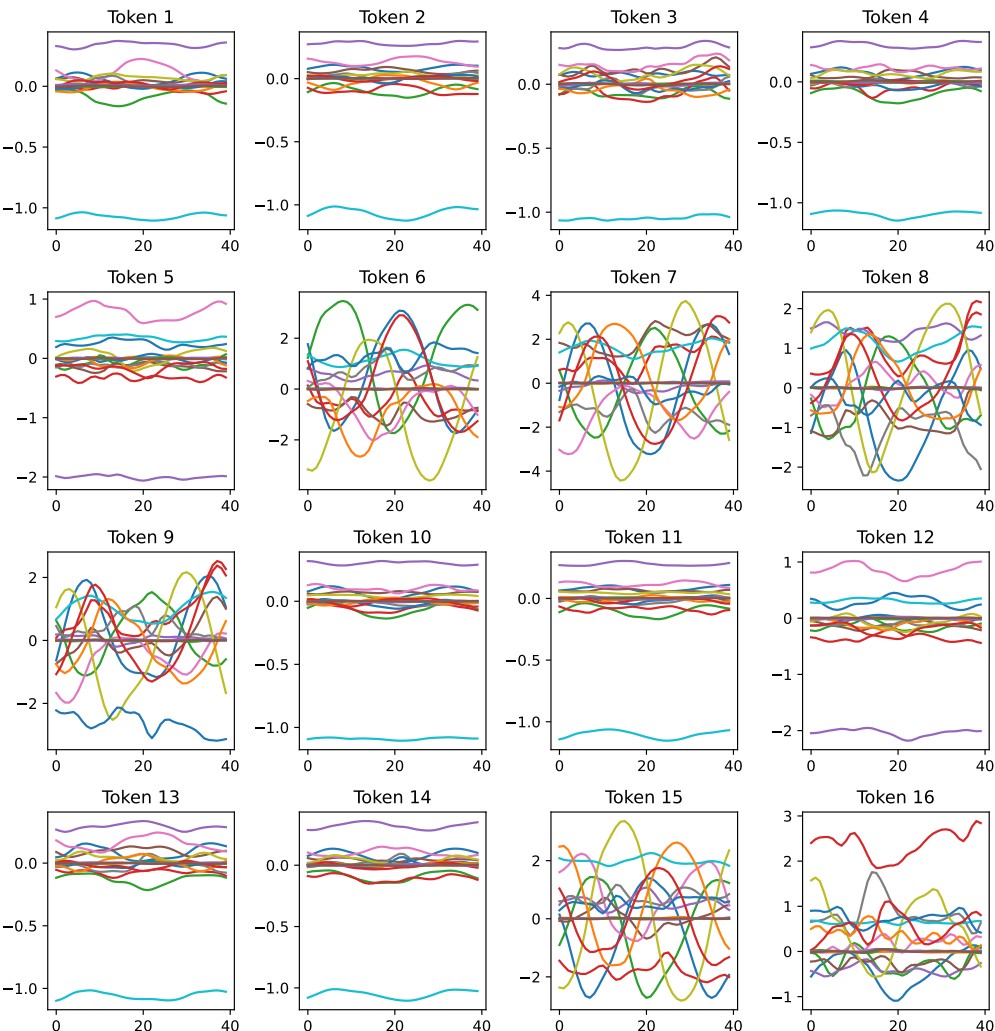

Figure 26: Latent space dynamics on *Navier-Stokes* - Predicted tokens over time. Each color line is a different token channel.

