# OpenReview forum: "AROMA: Preserving Spatial Structure for Latent PDE Modeling with Local Neural Fields"
_NeurIPS.cc/2024/Conference — NeurIPS 2024 poster_

### Official Review · Reviewer_qbYC · 2024-07-08

**Soundness:** 2
**Presentation:** 3
**Contribution:** 2
**Rating:** 6
**Confidence:** 4

**Summary:**

This paper presents a new pipeline for modeling PDE systems, especially for learning local neural fields. It designs a new encoder-decoder for absorbing any type of input, which avoids the constraint on meshes and cloud points. It can handle diverse geometries. A diffusion-based transformer architecture is used to model the latent dynamics. The numerical experiments have shown the effectiveness of the proposed method.

**Strengths:**

- This paper proposes an interesting encoder-decoder for handling arbitrary input geometries.

- This paper has done multiple experiments on different PDEs to compare the model performance.

- The paper is well-written and well-organized.

- This research topic is of interest to the general scientific machine-learning community.

**Weaknesses:**

- The motivation of using a VAE-type encoder-decoder is not well justified. The authors might add more details about the benefits of using a VAE setup for an encoder-decoder instead of a deterministic auto-encoder. Also, my concern is that using Gaussian distribution might constrain the representation capability of the latent features. The real-world dynamics or complex turbulence dynamics sometimes present heavy-tailed characteristics.

- This paper considers a diffusion-based transformer for learning latent dynamics. The training efficiency and computational memory might be an issue. Moreover, I think the clarification of its benefits is not sufficient. This module can be replaced by many other methods for modeling latent dynamics, such as NeuralODEs [1] or neural spectral methods [2]. It would be good to have an ablation study on the latent dynamics part.

**Refs:**

[1] Krishnapriyan, Aditi S., et al. "Learning continuous models for continuous physics." Communications Physics 6.1 (2023): 319.

[2] Wu, Haixu, et al. "Solving high-dimensional pdes with latent spectral models." arXiv preprint arXiv:2301.12664 (2023).

**Questions:**

- In lines 90-91, for the second stage of training, do you fix the encoder-decoder or pretrain encoder-decoder and then fine-tune the entire network?

- The authors claim that the encoder-decoder is principled. I didn’t see a principle or theoretical guarantee behind that.

- In Table 1, why does DNOT work worse than FNO on the 1D Burgers case? Also, as a standard baseline, FNO works pretty good among all baseline models and the proposed AROMA.

- I am also curious why FNO is not considered in the experiments of temporal extrapolation. FNO can also be modified in an auto-regressive way.

**Limitations:**

Please see my concerns in **Weaknesses** and **Questions**.

---

> ### Author Rebuttal · Authors · 2024-08-07
>
> * **W1** | The motivation of using a VAE-type encoder-decoder is not well justified.
>
> Thank you for your insightful comment. We chose to use a Variational Autoencoder (VAE) over a standard Autoencoder (AE) due to the VAE’s established reputation in the literature for producing compact latent representations while maintaining smoothness in the latent space. However, we acknowledge that there are various autoencoder types that employ different forms of regularization, and alternative methods could also be effective.
>
> We agree that enforcing regularization is crucial for obtaining a smooth latent space, whether through a variational approach, L2 regularization, or spectral regularization. Without such regularization, we observed that the latent space can exhibit increasing variance, leading to less stable results.
>
> In the supplementary pdf, Table 1 presents an ablation study comparing the VAE and AE across three datasets. This study demonstrates that an autoencoder with L2 regularization can be a viable alternative to the VAE for achieving a smooth latent space in some cases. The AE showed superior performance in terms of lower reconstruction errors for Burgers and Navier Stokes 1e-4, which translated into better rollout performance. However, for the more challenging Navier-Stokes 1e-5 dataset, the AE's latent space had high variance, which may account for the observed performance difference compared to the VAE.
>
> We appreciate your feedback, which enriches our analysis, and hope this explanation clarifies our approach and findings.
>
> * **W2** |  This paper considers a diffusion-based transformer for learning latent dynamics.
>
> Thank you for your feedback. In Table 1 of the supplementary material, we compare three versions of our model: AROMA with $K=3$ diffusion steps (as detailed in the paper), AROMA without diffusion, and a local MLP that disregards token relationships. As expected, the MLP, which ignores token interactions, performs poorly, while both the deterministic and diffusion transformers produce comparable results.
> It's important to note that the diffusion-based approach has the added advantage of generating a distribution, which opens the door to uncertainty modeling. This is particularly relevant given that the state-of-the-art in video prediction often relies on probabilistic models. To fully capture the benefits of diffusion models, it may be necessary to evaluate them using metrics beyond MSE, as these could better reflect their ability to model uncertainty. This is an area that warrants further exploration in our future work.
>
> Thank you for suggesting alternatives like NeuralODEs and neural spectral methods. We appreciate the opportunity to discuss our approach in this context.  We would like to emphasize that our chosen latent processor is particularly effective at capturing both local spatial information within tokens and global spatial information through token interactions, which is essential for accurately modeling global dynamics in our framework. In our study, the MLP is implemented with residual connections as described in [1], and while it has a structure similar to NeuralODEs, it lacks the ability to account for interactions between tokens, which is a key limitation.
>
> Similarly, the model proposed in [2] processes tokens independently at a given resolution using neural spectral blocks. Given this design, we expect that these models would exhibit comparable performance when applied to AROMA's latent tokens. However, the integration of token interactions in our approach is a critical factor that we believe enhances the modeling of complex dynamics.
>
> [1] Serrano et al. Operator Learning with Neural Fields: Tackling PDEs on General Geometries.
>
> [2] Krishnapriyan, Aditi S., et al. "Learning continuous models for continuous physics." Communications Physics 6.1 (2023): 319.
>
> * **Q1** |  In lines 90-91, for the second stage of training, do you fix the encoder-decoder ?
>
> Yes, we keep the encoder and decoder frozen during the second stage.
>
> * **Q2** |  The authors claim that the encoder-decoder is principled.
>
> We appreciate the feedback and agree that the wording may have been misleading. Rather than suggesting a theoretical guarantee, our intent was to emphasize that the encode-process-decode framework is a well-established and widely adopted approach in the research community. This method has proven effective in a variety of applications, and our use of it follows this standard practice.
>
> * **Q3** |  In Table 1, why does DNOT work worse than FNO on the 1D Burgers case?
>
> We took the recommended default parameters proposed in the publications and observed that GNOT was indeed less robust in dynamics modeling. In the 1D case, FNO has $\sim$ 4 million parameters while GNOT has about 0.8 million, this could explain the difference. As for FNO, we agree that it is a strong and robust baseline, however it is limited to regular grids and it cannot process point clouds as done here. Enhanced FNO versions have been developed for irregular meshes, but they are much less flexible than the method proposed here.
>
> * **Q4** | FNO can also be modified in an auto-regressive way.
>
> Indeed, prior research has explored and highlighted the temporal extrapolation capabilities of the Fourier Neural Operator (FNO). Given FNO's established performance, we anticipate that it will demonstrate comparable extrapolation effectiveness on our datasets as the AROMA model for regular grids only.
>
> In the first experimental section, we concentrated on dynamics over regular grids, where we implemented and trained FNO in an auto-regressive manner. In the second section, our focus shifted to irregular grids, where we aimed to compare our model against the most relevant baselines, specifically neural-field-based methods and transformers. To provide a more comprehensive evaluation, we plan to include the performance results of FNO and UNet on the regular-grid case $\pi = 100$ % in the camera-ready.

---

> > ### Comment · Reviewer_qbYC · 2024-08-12
> >
> > Thanks for your rebuttal. Some of my concerns have been addressed. I still have a question regarding the smoothness in the latent space. The Gaussian prior in VAE or L2 regularization in AE leads to smoothness. I think my previous concern has not been addressed, as listed below.
> >
> > > My concern is that using Gaussian distribution might constrain the representation capability of the latent features. The real-world dynamics or complex turbulence dynamics sometimes present heavy-tailed characteristics.
> >
> > This somehow explains why the Navier-Stokes 1e-5 dataset is more challenging in AE experiments of this paper.

---

> > > ### Author Response · Authors · 2024-08-12
> > > **Response on the regularization**
> > >
> > > Thank you for your answer.
> > >
> > > We have conducted additional experiments on the auto-encoder, specifically exploring your suggestion of not using any L2 regularization on the latent space for both Navier-Stokes 1e-4 and Navier-Stokes 1e-5 datasets. The results, expressed in relative L2 reconstruction errors on the test set (isolating the behavior of the encoder-decoder without involving dynamics), are shown in the following table:
> > >
> > > |                          | Navier Stokes 1e-4 | Navier Stokes 1e-5 |
> > > |--------------------------|--------------------|--------------------|
> > > | **With L2 Regularization**  | 7.30e-3            | 6.12e-2            |
> > > | **Without Regularization** | 3.49e-2            | 7.66e-2            |
> > >
> > > We observed that without L2 regularization, the training process led to a much higher variance in the latent space that with the regularization (with standard deviations exceeding 100). This increased variance is likely a key factor contributing to the higher reconstruction errors observed on the test set, as well as the noticeable slowdown in the training process.
> > >
> > > Previous works have successfully used a CNN VAE to obtain latent reduced representations in the context e.g. of precipitation nowcasting [1]. However, you are correct in suggesting that other forms of regularization might be more suitable for capturing the complexity of turbulence dynamics. Still, the L2 regularization serves two key roles in our approach:
> > >
> > > *  **Facilitating the training of the encoder-decoder** by keeping the latent codes within a manageable range, which helps in maintaining the stability and efficiency of the learning process.
> > > * **Facilitating the training of the transformer**, which benefits from a more controlled and predictable latent space.
> > >
> > > While L2 regularization might not be the optimal choice for all cases, it plays a crucial role in ensuring that the model remains trainable and that the latent space is not excessively scattered.
> > >
> > > We appreciate your suggestion and agree that exploring alternative regularization techniques could yield further improvements, especially in the context of capturing heavy-tailed distributions in complex datasets.
> > >
> > > [1] Gao et al. PreDiff: Precipitation Nowcasting with Latent Diffusion Models. Neurips 2023.

---

> > > > ### Comment · Reviewer_qbYC · 2024-08-12
> > > >
> > > > Thanks for your quick response. That already looks good to me. I will raise my score.

---

> > > > > ### Author Response · Authors · 2024-08-12
> > > > > **Response**
> > > > >
> > > > > Thank you for your thoughtful remarks and detailed review. We greatly appreciate the time and effort you put into providing valuable insights regarding the different blocks of our architecture. We will certainly explore ways to better represent real-world and highly turbulent dynamics in our future work.

---

### Official Review · Reviewer_xV3P · 2024-07-12

**Soundness:** 3
**Presentation:** 3
**Contribution:** 2
**Rating:** 7
**Confidence:** 4

**Summary:**

The paper proposes a framework using autoencoder and diffusion transformer for predicting the forward dynamics of time-dependent PDEs. Leveraging cross-attention and neural fields, the framework is able to handle different types of meshes and geometries. The authors demonstrate the effectiveness of their proposed framework on several 1D and 2D PDE problems with different geometries.

**Strengths:**

1. The proposed framework adopted many existing techniques in a natural and reasonable way, such as using local neural fields and cross attention to pass information between nodes in different meshes, diffusion-based predictor/refiner. The motivation and benefits of various components in the framework are illustrated clearly.

2. The empirical performance of the proposed model is strong across several benchmarks.

3. The authors explain their method well and overall the paper is easy to follow.

**Weaknesses:**

1. Many of the techniques used in the paper are taken from existing models (authors have properly acknowledged them), so the technical advancement for the paper might not be that much. Nevertheless, I think the authors have done a decent job tweaking them to work well, evidenced by the experiments, and explained why these techniques can be helpful to PDE modeling.

2. One issue in the experiment is that the authors have employed a diffusion predictor to predict the dynamics (which can be seen as a predictor-corrector scheme) whereas other baselines are deterministic single-step predictor. With that said, the proposed framework uses more NFE (proportional to the number of diffusion steps) than other baselines. The authors did provide an ablation on Burgers' which shows that AROMA without diffusion still outperforms other model, but it is unclear for other more chaotic and higher-dimensional cases.

**Questions:**

1. For systems that feature a decaying spectrum (e.g. fluid flow with relatively high viscosity), the proposed method is a good fit, which is not surprising. Will the method still perform well on slightly more chaotic systems like KS-equation or 2D Komogorov flow and how does the spectrum of reconstructed snapshots from decoder look like in these cases?

2. How many refinement steps does the model use?

3. There are some closed related works that not have been discussed. For example, the similar idea of  learnable latent tokens have been explored in "Solving high-dimensional pdes with latent spectral models." Using pretrained autoencoder to dervie a mesh-reduced space and then learn to forecast dynamics in the latent space is also studied in "Predicting Physics in Mesh-reduced Space with Temporal Attention" and "Latent Neural PDE Solver: a reduced-order modelling framework for partial differential equations".

4. Typos:
* line 228: 3D Shallow-Water Equation (Navier-Stokes1 × 10−3)
* Table 4&5: "opochs"

**Limitations:**

The authors have discussed the limitation in the conclusion part.

---

> ### Author Rebuttal · Authors · 2024-08-07
>
> * **W1**  | Many of the techniques used in the paper are taken from existing models [...] .
>
> Thank you for your feedback. It is true that our paper builds upon established techniques and models, which we have duly acknowledged. However, our contributions lie not only in leveraging these techniques but in integrating and adapting them in a novel way to advance PDE modeling. Specifically:
>
> 1. *Novel Integration*: We present a unique framework that combines attention blocks and neural fields in a way that has not been done before, particularly for processing domain geometries and unrolling dynamics in PDEs. This integration allows for a more streamlined and effective approach to handling complex geometries and dynamics. Up to our knowledge, this is the first approach that allows for an automatic implicit latent encoding, leveraging local and global spatial relations, for a diversity of geometries, for modeling spatio-temporal dynamics. Concurrent approaches, [1,2] make use of pre-defined patching strategies that are less flexible for handling these diverse geometries.
>
> 2. *Empirical Validation*: The experiments we conducted demonstrate that our approach achieves state-of-the-art performance across various datasets, confirming the effectiveness of our formulation. More importantly, these results validate our hypothesis that a latent space preserving spatial information is crucial for accurately modeling dynamics in a reduced representation. Additionally, our method systematically outperforms other transformer architectures, underscoring its robust performance and well-founded approach.
>
> 3. *Insights and Rationale*: We experimentally validate through cross-attention and perturbations that our latent space has a spatial interpretation.
>
> [1] Solving High-Dimensional PDEs with Latent Spectral Models, Wu et al 2023.
> [2]  Universal Physics Transformers. Alkin et al. (2024).
>
> * **W2**. | One issue in the experiment is that the authors have employed a diffusion predictor [...] .
>
> In the supplementary PDF, we present an ablation study showing that diffusion is not essential for accurate predictions, as AROMA without diffusion still outperforms Neural Field and Transformer baselines (Table 1). However, diffusion improves long rollouts, as seen in Figure 3 (Burgers) and Figure 1b (KS equation).
>
> The diffusion approach opens the door to uncertainty modeling, and is state-of-the-art for e.g. video prediction. To fully leverage this for PDE, future work should consider metrics beyond MSE to better capture the advantages of diffusion models.
>
> * **W3** | With that said, the proposed framework uses more NFE [...].
>
> First, using a small number of diffusion steps $(K=3)$ strikes a good balance between accuracy and computational cost, as the processor works with fewer tokens ($M < N$).  Table 1 in the supplementary pdf compares three model versions: AROMA with $(K=3)$ steps, AROMA without diffusion, and a local MLP.  The MLP which ignores the relation between tokens does not perform well, while deterministic and diffusion transformers show similar results.  Finally, although using few diffusion steps helps with noise augmentation and for long rollouts, it does not fully utilize diffusion’s potential, which typically requires more steps (e.g., K=1000). For datasets characterized by chaotic dynamics, future work could explore training with a full diffusion process ($K=1000$).
>
> * **Q1** | For systems that feature a decaying spectrum [...].
>
> This is an excellent question. As you correctly point out, the encoder-decoder architecture may indeed become a bottleneck in scenarios where high-frequency amplitudes are challenging to model and represent accurately.
>
> To investigate this point, we conducted an experiment with the KS equation using the specific setup described by [3]. In this setup, the model predicts $u_{t+4\Delta t}$ from  $u_t$ over training trajectories spanning $[0, 140 \Delta t]$, and during testing, predictions are unrolled up to  $[0, 640 \Delta t]$. Note that we do not employ the "predict-difference" trick here. In the dataset, each trajectory is generated with varying grid resolutions ($\Delta x$) and time-stepping intervals ($\Delta t$). To accommodate these variations, we provided additional tokens $T_{\Delta x}$  and $T_{\Delta t}$ as context for the transformer, while keeping the rest of the architecture unchanged. The results are detailed in Table 2 and Figure 1.
>
> Interestingly, while the deterministic version of the model achieves a lower 1-step prediction mean squared error (MSE) compared to the diffusion version, the diffusion model shows better performance in maintaining high correlation over longer time horizons. The primary identified limitation is the reconstruction capability of the encoder-decoder, which impacts the overall predictive accuracy of the transformer. Figure 1.a in the pdf page, represents the amplitude spectrum of AROMA's prediction vs the real spectrum. We can see in this figure that AROMA's decoder overestimates the weights of high frequencies and is faithful up to the 40th mode. In order to remedy to this problem, one would need to lower the reconstruction error corresponding to these frequencies. Specifically, AROMA achieved an MSE reconstruction of $1 \times 10^{-6}$ (i.e. a Relative L2 error of $0.08$%) on this dataset, when a relative error around $0.001$% would be optimal. This is a non trivial issue requiring further investigations.
>
> [3] Lippe, et al. PDE-Refiner: Achieving Accurate Long Rollouts with Neural PDE Solvers. Neurips 2023.
>
> * **Q2** | How many refinement steps does the model use?
>
> We use K=3 steps throughout the paper for all experiments. We chose this number based on the study of Lippe et al. 2023 and some preliminary trials.
>
> * **Q3** | There are some closed related works that not have been discussed.
>
> Thank you for these references. We will add them together with a discussion to the related work section for the camera-ready.

---

> > ### Comment · Reviewer_xV3P · 2024-08-13
> > **Reply to authors' rebuttal**
> >
> > Thanks for the response and update. The majority of the questions and concerns I raised have been addressed and I have adjust my rating accordingly.

---

> > > ### Author Response · Authors · 2024-08-13
> > > **Response**
> > >
> > > Thank you for your detailed feedback. Your comments have significantly contributed to enhancing the quality and clarity of our manuscript.

---

### Official Review · Reviewer_gPBs · 2024-07-13

**Soundness:** 3
**Presentation:** 3
**Contribution:** 3
**Rating:** 5
**Confidence:** 5

**Summary:**

The paper introduces a novel framework designed to enhance the modeling of partial differential equations (PDEs) using local neural fields. It proposes a flexible encoder-decoder architecture that achieves smooth latent representations of spatial physical fields from various data types and employs a diffusion-based formulation to achieve greater stability and enable longer rollouts compared to conventional MSE training. The authors show the superior performance of their framework on 1D and 2D PDEs.

**Strengths:**

The authors claim that the proposed method can handle a variety of data types including regular-grid inputs and point clouds, eliminating the need for patching and allowing efficient processing of diverse geometries.

The diffusion-based formulation enhances stability and enables longer rollouts compared to traditional methods.

AROMA demonstrates superior performance in simulating 1D and 2D equations, capturing complex dynamical behaviors effectively.

The framework leverages attention blocks and neural fields, resulting in a model that is easy to train and achieves state-of-the-art results without requiring prior feature engineering.

**Weaknesses:**

Although the proposed method reduces computational cost compared to existing transformer architectures, it may still be computationally expensive for very large datasets or extremely high-resolution simulations.

The experiments are performed on benchmark datasets, and the performance on larger and real-world examples remains to be demonstrated.

It is also unclear if the approach always produces stable results.

**Questions:**

It is unclear what the authors mean by patching for data input types.
The visual results for the cylinder and aerofoil cases are not provided. Is there any specific reason these were not included?
What would be the training cost of extending this to 3D PDEs.

**Limitations:**

While AROMA demonstrates effective performance on small-size datasets, its scalability to larger datasets and more complex real-world scenarios is not fully established. Scaling up neural field-based methods to handle larger volumes of data without compromising performance remains a challenge.

---

> ### Author Rebuttal · Authors · 2024-08-07
>
> * **W1** | Although the proposed method reduces computational cost,  [...] .
>
> Our architecture employs an encoder-decoder structure that includes cross-attention blocks. The computational complexity of these cross-attention operations is $\mathcal{O}(NMd)$, where $N$ represents the number of observations, $M$ is the number of latent tokens, and $d$ is the hidden dimension. This gives our model a linear time complexity of $\mathcal{O}(N \times K)$ with respect to the number of observations, where $K$ is a constant factor. Note that $M$ and $d$ ($K=Md$) are parameters of the architecture that could be set at different values, allowing for a compromise between accuracy and complexity.
>
> We acknowledge that, despite this linear complexity, the constant $K$ can still be substantial when dealing with large datasets or high-resolution simulations. This highlights a trade-off between computational efficiency and the scale of the data being processed. While our approach significantly reduces the computational cost compared to traditional transformer architectures, we recognize that further optimization or alternative strategies may be needed to handle extremely large datasets or high-resolution scenarios effectively.
>
> * **W2** | The experiments are performed on benchmark datasets, [...].
>
> The current experiments utilize benchmark datasets to establish baseline performance and validate the effectiveness of the proposed method in a controlled setting. These datasets, introduced in various studies (Pfaff et al. 2020, Li et al. 2020, Yin et al. 2022, Brandstetter et al. 2022), were generated using different solvers and for different objectives. AROMA has demonstrated consistent performance across all these datasets, highlighting its robustness. Nevertheless, it is crucial to evaluate its applicability and performance in larger, real-world scenarios. Future work will focus on scaling the method to address larger datasets and more complex, real-world applications. In order to better assess the capabilities of AROMA, we have performed additional experiments on the more challenging and chaotic KS-equation (see Table 2 and Figure 1 of the pdf page) - see also the comments in the "global response" to the reviewers. In conclusion, when AROMA performs extremely well on a large range of dynamics and problems, like all the models using a reduced latent representation, it faces difficulties for more complex problems involving chaotic phenomena that require more specific approaches.
>
>
> * **W3** | It is also unclear if the approach always produces stable results.
>
> Our results provide strong evidence of the method's stability. Notably, the time-correlation plot in Figure 3 indicates that AROMA maintains stability during extended rollouts on Burgers' equation. Additionally, Table 2 demonstrates that the method can effectively extrapolate beyond the training horizon and remains stable under perturbations of the observations.
>
> * **Q1** | It is unclear what the authors mean by patching for data input types.
>
> The term comes from visual transformers operating on regular grids, and in this case refers to a predefined regular spatial partition of the point grids. In a more general context, the term "patching" refers to a process that can be somewhat arbitrary, particularly when dealing with irregular meshes. In such cases, patching should ideally be derived from a systematic partitioning of the data. For example, [1] employs this idea by partitioning the domain on different resolutions to obtain frame patches. [2] considers graph representations of input points and agglomerates points into "supernodes" that are then embedded in a latent space. In contrast, AROMA eliminates the need for transforming data into patches. Instead, it directly learns a spatial representation of the data, making it more flexible and effective across various data structures without relying on predefined patches.
>
> [1] Solving High-Dimensional PDEs with Latent Spectral Models, Wu et al 2023.
> [2] Alkin, B., Fürst, A., Schmid, S., Gruber, L., Holzleitner, M., & Brandstetter, J. (2024). Universal Physics Transformers. 1–27.
>
> * **Q2** | The visual results for the cylinder and aerofoil cases are not provided.
>
> Thank you for this remark. We have added visualizations of AROMA's predictions in the rebuttal pdf page for the dataset Cylinder in the *Out-t* regime. We will provide additional visualizations of the results for the camera ready.
>
> * **Q3** | What would be the training cost of extending this to 3D PDEs.
>
> As detailed in our answer to **W1**, our model exhibits a linear complexity of $\mathcal{O}(NMd)$, where $N$ is the number of observations, $M$ is the number of latent tokens, and $d$ is the hidden dimension. For a domain $\Omega$ discretized on a regular 3D grid of size $L$, the total number of observations is $N = L^3$. We have not conducted 3D experiments with AROMA yet, and therefore do not know the number $M$ of tokens to allow for a faithful reconstruction and dynamics modeling. We then provide some indications below by considering two scenarios.
>
> If we apply a downsampling factor of 4 to determine the number of tokens (as done in the N.S. experiments in the paper), then the number of tokens $M$ becomes $(L/4)^3$. Hence, the additional complexity due to scaling from 2D to 3D, with downsampling, is proportional to $L^2 / 4$. For example, with $L = 32$, the computational cost in 3D is multiplied by $256$ compared to 2D.
>
> If we maintain the number of tokens fixed (i.e., not using a downsampling rule), the complexity simply scales with the increase in dimensions, resulting in a cost multiplication factor of $32$ when moving from 2D to 3D.
>
> In future work, we plan to experimentally investigate how AROMA scales with these two strategies when moving from 2D to 3D.

---

> > ### Comment · Reviewer_gPBs · 2024-08-12
> >
> > Thank you for the detailed response.

---

### Official Review · Reviewer_rnRB · 2024-07-16

**Soundness:** 4
**Presentation:** 4
**Contribution:** 4
**Rating:** 9
**Confidence:** 5

**Summary:**

An innovative approach for improving the modeling of partial differential equations (PDEs) using local neural fields is presented in the paper "AROMA: Preserving Spatial Structure for Latent PDE Modeling with Local Neural Fields" (Attentive Reduced Order Model with Attention). Through the provision of a versatile encoder-decoder architecture that can handle a variety of data sources, including regular grids and point clouds, AROMA seeks to alleviate the shortcomings of current neural operator models.

**Strengths:**

na

**Weaknesses:**

na

**Questions:**

1. What are the main drawbacks that AROMA seeks to solve with respect to current neural operator models for PDEs?
3. What function does the AROMA architecture's conditional transformer serve?
5. How can AROMA guarantee the accuracy and stability of its forecasting models?
5. How is the model's performance improved by the encoder and decoder's two-stage training process?
6. In the experiments, what kinds of datasets were employed, and how did AROMA fare in comparison to other models?

---

> ### Author Rebuttal · Authors · 2024-08-07
>
> * **Q1** | What are the main drawbacks that AROMA seeks to solve with respect to current neural operator models for PDEs?
>
> AROMA addresses key limitations of existing neural operator models for PDEs. Traditional transformer-based methods, such as those by Li et al. (2023) and Hao et al. (2023), unroll dynamics directly in the original space, leading to high complexity and inefficiency. Neural-field methods such as DINO or CORAL unroll the dynamics in a latent space without spatial prior. AROMA mitigates these issues by encapsulating domain geometry and observation values into a compact latent representation, allowing for efficient forecasting with reduced computational cost. This approach simplifies the training process and avoids the need for prior feature engineering, making it particularly effective for complex geometries.
>
>
> * **Q2** | What function does the AROMA architecture's conditional transformer serve?
>
> In AROMA, the conditional transformer plays a crucial role in modeling and forecasting dynamics efficiently. It operates on a fixed-size compact latent token space that encodes local spatial information. This transformer processes the encoded spatial data and models the dynamics while capturing spatial relations both locally and globally across tokens. Additionally, the conditional neural field utilized in the decoding stage enables querying forecast values at any point within the spatial domain, enhancing the model's flexibility and accuracy. Said otherwise, AROMA is a mesh free solution that accepts any geometry as input and can be queried at any point in the spatial domain.
>
>
> * **Q3** | How can AROMA guarantee the accuracy and stability of its forecasting models?
>
> As AROMA is entirely data-driven, there is no absolute guarantee that the model will always approximate the true solution perfectly at inference. However, extensive experiments have been conducted throughout the paper to assess the accuracy and stability of the method under various initial conditions. Experiments have been performed on long rollouts (see appendix C.2 Fig. 9 for example), including forecasting beyond the training horizon domain. The results consistently align with state-of-the-art methods, suggesting that AROMA is stable for all the equations studied. Additionally, by incorporating a diffusion transformer to unroll the dynamics, it is possible to generate multiple solutions and analyze their variance or the average cross-correlation between these solutions. Said otherwise it allows to generate samples from the predictive distribution of the trajectories. This approach allows computing different statistics on this distribution, and helps in identifying when the model may have diverged from the expected ground truth. Preliminary results illustrating this behavior are presented in Figure 10 of the paper.
>
>
> * **Q4** | How is the model's performance improved by the encoder and decoder's two-stage training process?
>
> The two-stage training process in AROMA is very stable and aligns with previous works in the literature. This separation allows for specialized training of each component, ensuring that the encoder-decoder effectively encodes spatial features and the processor accurately predicts dynamics. Experimentally, it is easier to train than the end-to-end training alternative.
>
>
> * **Q5** | In the experiments, what kinds of datasets were employed, and how did AROMA fare in comparison to other models?
>
> The experiments were conducted on representative spatio-temporal forecasting problems, including 1D and 2D dynamics with periodic boundary conditions, as well as domains with complex geometries and very diverse input types, such as point sets, grids, and meshes. AROMA exhibited outstanding performance on these datasets, achieving state-of-the-art results when compared to existing neural field and transformer-based methods.

---

### Author Rebuttal · Authors · 2024-08-07

Dear Reviewers,

Thank you for your insightful feedback.

In response to your comments, we have addressed the key concerns by providing additional experimental results. Please refer to the supplementary PDF, specifically:

* An ablation study in Table 1 that compares different processing blocks  (Diffusion (K=3 steps) vs deterministic Transformer vs MLP) for AROMA on Burgers, Navier-Stokes 1e-4, and Navier Stokes 1e-5. The deterministic transformer and diffusion transformer share the same architecture, while the MLP is implemented with residual connections as in [1].
* An ablation study, also in Table 1, on the impact of the choice of encoder and decoder framework (Autoencoder with L2 regularization vs VAE) for AROMA on Burgers, Navier-Stokes 1e-4, and Navier Stokes 1e-5.
* Additional results on KS equation in Table 2 and Figure 1 of the pdf page, with a comparison between the diffusion and deterministic versions of AROMA.
* Visual comparisons between AROMA’s predictions and the ground truth on the Cylinder Flow dataset in Figure 2 of the pdf page.

From these new results, we can draw the following conclusions:

* Processing Blocks: Modeling  interactions at the local and global levels is key to learn the dynamics faithfully. Experiments using MLPs (table 1) as time steppers which do not consider interactions between tokens lead to significantly lower performance compared to transformers.

* Deterministic vs. Diffusion: The deterministic version of AROMA shows consistently robust performance and even surpasses the diffusion version on the Navier-Stokes 1e-4 case. This demonstrates that the latent tokens obtained with AROMA contain meaningful information for dynamics modeling. On the other hand, the deterministic version yields less accurate long rollouts on Burgers or KS. Note that using diffusion allows us to model the trajectory distribution, which opens the way to infer statistics on this distribution. This is key, for example, when modeling uncertainty, which is a critical problem for these models.

* Encoder-Decoder Frameworks: Using an autoencoder with L2 regularization is a viable alternative to the VAE for achieving a smooth latent space. The autoencoder demonstrated superior performance on two datasets, explained by its lower reconstruction errors, which translate into better rollout performance. However, for the more challenging Navier-Stokes 1e-5 case, the autoencoder's latent space exhibited high variance, which may explain the observed performance difference with the VAE.
* KS Equation Limitations: AROMA currently struggles with dynamics that exhibit chaotic phenomena and non-decaying spectra, as shown by the KS equation results. The primary limitation appears to be the reconstruction capabilities of the encoder-decoder. For the KS equation, we found that obtaining reconstructions with an MSE in the range of 1e-10 to 1e-12 was necessary for accurate spectrum reconstruction. Like all models leveraging a reduced latent representation space, AROMA inherently lose some of the fine-grained details necessary for accurately capturing chaotic behavior. In conclusion, while AROMA performs very well on simpler dynamics, dealing with chaotic phenomena requires more involved modeling that explicitly targets the chaotic component. Note that using dedicated modules for this purpose is a current practice in fluid dynamics - e.g. LES (Large eddy simulation).

We hope these additional results and insights address your concerns and enhance the understanding of AROMA’s capabilities and limitations.

[1] Serrano et al. Operator Learning with Neural Fields: Tackling PDEs on General Geometries.

---

### Decision · Program_Chairs · 2024-09-25

**Decision:**

Accept (poster)

**Comment:**

This paper received initially somewhat mixed reviewed.  The authors submitted a rebuttal with some additional experimental results and there was an active discussion period with many of the issues noted in the initial reviews being addressed.  After discussion the reviewers are now consistently positive.

While there was some discussion around novelty and scalability, all reviewers believed the work warranted publication and the AC agrees.